# An in vitro model for vitamin A transport across the human blood–brain barrier

Chandler B Est[†], Regina M Murphy*

Department of Chemical and Biological Engineering, University of Wisconsin, Madison, United States

**Abstract** Vitamin A, supplied by the diet, is critical for brain health, but little is known about its delivery across the blood–brain barrier (BBB). Brain microvascular endothelial-like cells (BMECs) differentiated from human-derived induced pluripotent stem cells (iPSCs) form a tight barrier that recapitulates many of the properties of the human BBB. We paired iPSC-derived BMECs with recombinant vitamin A serum transport proteins, retinol-binding protein (RBP), and transthyretin (TTR), to create an in vitro model for the study of vitamin A (retinol) delivery across the human BBB. iPSC-derived BMECs display a strong barrier phenotype, express key vitamin A metabolism markers, and can be used for quantitative modeling of retinol accumulation and permeation. Manipulation of retinol, RBP, and TTR concentrations, and the use of mutant RBP and TTR, yielded novel insights into the patterns of retinol accumulation in, and permeation across, the BBB. The results described herein provide a platform for deeper exploration of the regulatory mechanisms of retinol trafficking to the human brain.

## eLife assessment

This **fundamental** work substantially advances our understanding of retinol transport through the blood–brain barrier. The evidence supporting the conclusions is **compelling**, with rigorous biochemical assays. In general, the work is of broad interest to cell biologists, biochemists, and neuroscientists.

*For correspondence:
regina.murphy@wisc.edu

Present address: [†]Division of Endocrinology, Metabolism and Lipid Research, Washington University School of Medicine, 660 South Euclid Avenue, St. Louis, United States

Competing interest: The authors declare that no competing interests exist.

## Introduction

Retinoids (vitamin A and related compounds) are essential micronutrients supplied by the diet that regulate over 500 genes (*Balmer and Blomhoff, 2002*) and are involved in vision, embryonic development, cell differentiation, metabolism, and brain health (*Wald, 1934*; *Clagett-Dame and Knutson, 2011*; *Love and Gudas, 1994*; *Shearer et al., 2012*). Retinoids play critical roles in both brain development and maintenance of brain health. For example, retinoic acid is a key signaling compound that induces neural differentiation and acquisition of unique brain vasculature properties in the prenatal brain (*Maden, 2007*; *Bonney et al., 2018*). In the mature brain, retinoids contribute to maintenance of synaptic plasticity and sleep regulation; vitamin A deficiencies can lead to learning and memory deficits and depression of long-term potentiation (*Aoto et al., 2008*; *Lane and Bailey, 2005*; *Ransom et al., 2014*; *Cocco et al., 2002*). Vitamin A levels (circulating principally as retinol) naturally diminish with time, and there is mounting evidence that patients with Alzheimer's disease (AD) have lower serum levels of vitamin A than age-matched controls (*Jiménez-Jiménez et al., 1999*; *Zaman et al., 1992*), and that retinoid deficiencies contribute to cognitive decline in AD (*Jiménez-Jiménez et al., 1999*; *Goodman and Pardee, 2003*; *Zeng et al., 2017*). Retinoids regulate expression of several genes that have been implicated in AD (*Goodman and Pardee, 2003*), specifically those genes involved in generation (*Yang et al., 1998*; *Lahiri and Nall, 1995*; *Fukuchi et al., 1992*; *Wyss-Coray*

*et al., 2001*; *Tippmann et al., 2009*; *Wang et al., 2015*; *Culvenor et al., 2000*) and clearance (*Zhao et al., 2014*; *Melino et al., 1996*; *Goncalves et al., 2013*) of the AD-related peptide beta-amyloid. Reciprocally, AD pathology may disrupt normal vitamin A trafficking (*Goncalves et al., 2013*; *Boerwinkle et al., 1994*).

The mobilization of retinol (ROH) stored in the liver is well characterized. Briefly, ROH is packaged with retinol-binding protein 4 (RBP) (*Zanotti and Berni, 2004*) in hepatocytes and then secreted into the blood. There, the ROH-RBP complex binds a second protein, transthyretin (TTR), thereby preventing clearance of the complex in the kidney (*Monaco, 2000*). The distribution of ROH across its primary protein transporters in the blood is shown in *Figure 1A*.

In order to exert biological activity, ROH must cross from the blood into target tissues. There remain several unsettled questions regarding the mechanisms of vitamin A uptake (see *Zhong et al., 2014*). ROH is lipophilic, and several in vitro studies demonstrate that 'free' ROH (not bound to RBP) can enter lipid vesicles and cross bilayers through diffusion (*Noy and Xu, 1990b*; *Noy and Xu, 1990a*; *Fex and Johannesson, 1990*; *Fex and Johannesson, 1988*). Due to its lipophilicity, ROH binds serum albumins non-specifically, with $K_D$'s reported between 200 and 7600 nM (*N'soukpoé-Kossi et al., 2007*; *Belatik et al., 2012*); however, in humans serum, albumins are not believed to transport ROH in vivo (*Krinsky et al., 1958*), nor participate directly in the transport of ROH across the cell membrane. Other studies provide evidence that cell-surface protein(s) participate in RBP-mediated ROH transport across cell membranes (*Chen and Heller, 1977*; *Rask and Peterson, 1976*; *McGuire et al., 1981*; *Pfeffer et al., 1986*), of which one has been identified as the transmembrane protein STRA6 (signaling receptor and transporter of retinol) (*Kawaguchi et al., 2007*). STRA6 binds RBP, mediates ATP-independent bidirectional transfer of ROH (*Isken et al., 2008*; *Kawaguchi et al., 2012*; *Kawaguchi et al., 2013*; *Muenzner et al., 2013*), and is involved in cell signaling (*Berry et al., 2011*; *Berry et al., 2012b*; *Chen et al., 2012*). It is well established that ROH must be released from RBP to enter the cell via STRA6 and that RBP is not internalized (*Chen and Heller, 1977*; *Rask and Peterson, 1976*). ROH uptake is enhanced through coupling with cellular retinol binding protein 1 (CRBP1), the primary intracellular ROH binding protein in most tissues (*Kawaguchi et al., 2013*; *Kawaguchi et al., 2011*), and lecithin retinol acyltransferase (LRAT), the enzyme chiefly responsible for esterification of ROH into retinyl esters (RE), the 'long-term storage' form of vitamin A (*Kawaguchi et al., 2007*; *Isken et al., 2008*). Recent cryo-EM characterization of STRA6 provides structural evidence that STRA6 may mediate ROH transfer from both free lipid and/or ROH-RBP complexes (*Chen et al., 2016*), but no functional data were provided. STRA6 can mediate transfer of ROH from preparations of ROH-RBP-TTR (*Kawaguchi et al., 2007*; *Kawaguchi et al., 2011*; *Berry et al., 2012a*); however, ROH-RBP-TTR does not appear to directly bind cell surfaces (*Berry et al., 2012a*), which suggests dissociation of ROH-RBP complex from TTR is required prior to ROH delivery.

The blood–brain barrier (BBB) is the major site of nutrient exchange between the brain and circulation (*Daneman and Prat, 2015*). The primary barrier phenotype is imparted by brain microvascular endothelial cells, which are heavily polarized: the luminal (blood-facing) and abluminal (brain-facing) membranes differ significantly in lipid and protein composition (*Worzfeld and Schwaninger, 2016*). The mechanism(s) by which vitamin A is accumulated at the luminal membrane and then permeated across the abluminal membrane remains largely unexplored, in large part due to the inherent difficulty in conducting controlled in vivo studies. A diagram of suspected ROH delivery modes across the BBB is shown in *Figure 1B*.

Investigation into the mechanism of ROH transport to the brain would be greatly facilitated by well-characterized in vitro models of the human BBB. Recently, techniques have been developed for reprogramming human-derived cells into an induced pluripotent state (iPSCs), which can be differentiated into cell types that are difficult to obtain from primary tissues (*Yu et al., 2007*; *Takahashi and Yamanaka, 2006*). Of relevance to this work, iPSCs have been differentiated into brain microvascular endothelial-like cells (BMECs) (*Lippmann et al., 2014*; *Lippmann et al., 2012*). These iPSC-derived BMECs express many markers of in vivo BMECs, and they are robust and readily produced. Critically, iPSC-derived BMECs show transcriptional expression of STRA6 (*Lippmann et al., 2012*), suggesting that vitamin A uptake at the luminal BBB membrane may be facilitated by STRA6 in a similar manner to other cell types. However, the mechanism of vitamin A permeation at the BBB abluminal membrane remains wholly unexplored.

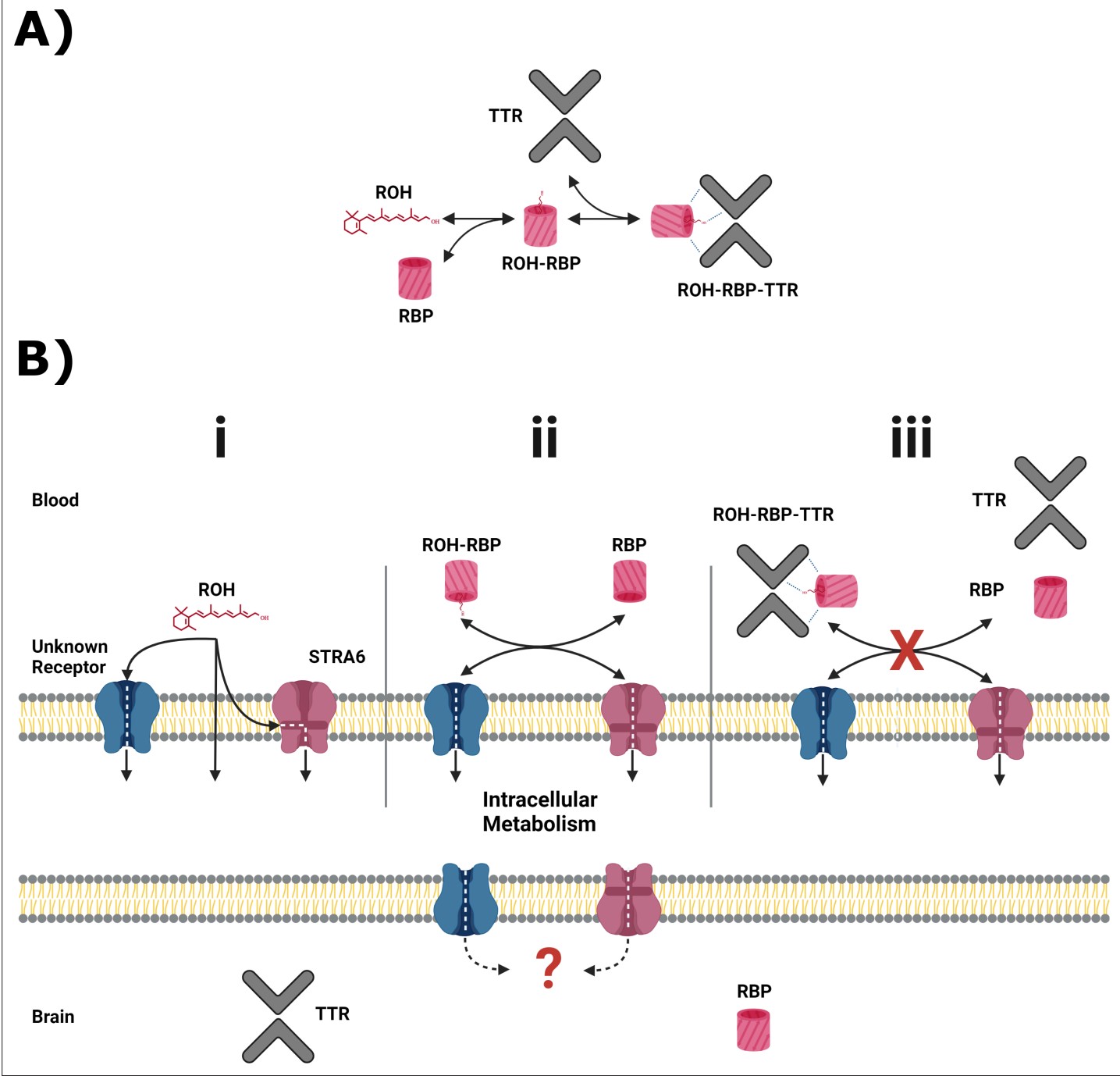

**Figure 1.** Primary serum vitamin A carriers and potential routes of delivery at the blood–brain barrier (BBB). (**A**) Principal retinol distribution between free-lipid and protein-bound states in the blood. Retinol (ROH) in the blood partitions between free-lipid and protein-complexed states, with retinol-binding protein (RBP) and transthyretin (TTR) serving as the principal serum ROH transporters. (**B**) Putative ROH delivery mechanisms at the BBB. Critically, RBP and TTR do not cross the BBB. (**i**) In the blood, free ROH may cross the BBB by lipophilic diffusion through the lipid bilayer or through a specific cell-surface transporter, such as STRA6. (**ii**) ROH in complex with RBP (ROH-RBP) is known to deliver ROH to cell-surface transporters, including STRA6. STRA6 is thought to mediate exchange of retinoids between blood and intracellular pools. The net accumulation or release of ROH is thereby dependent on the ratio of ROH-RBP and unbound RBP present in the blood. (**iii**) ROH-RBP is complexed with TTR as it circulates in the blood. The ROH-RBP-TTR complex has not been shown to directly bind to cell-surface transporters, including STRA6, suggesting that dissociation of ROH-RBP from TTR is required before ROH is internalized as shown in panel (**ii**). Regardless of the entry method, all ROH is thought to enter the intracellular retinoid metabolism. It is unknown how intracellular retinoids are transported into the brain. RBP and TTR are expressed in the brain and presumably transport ROH throughout the brain.

**Table 1.** ROH and protein concentrations for BMEC accumulation and permeability assays.

ROH distribution based on $K_D$ (µM)*

| Total ROH concentration (µM) | Delivery mode† | Free ROH | ROH-RBP | ROH-RBP-TTR | Notes |
|---|---|---|---|---|---|
| 0.1 | Free | 0.1 | 0 | 0 | |
| 0.4 | Free | 0.4 | 0 | 0 | |
| 2 | Free | 2 | 0 | 0 | |
| 2 | ROH-RBP | 0.4 | 1.6 | 0 | ROH partitions between free and RBP-bound states. |
| 2 | ROH-muRBP | 0.4 | 1.6 | 0 | muRBP binding affinity to TTR and possibly to STRA6 is abolished. |
| 2 | ROH-RBP-TTR | 0.14 | 0.18 | 1.68 | ROH partitions between free, RBP-bound, and RBP-TTR-bound states. |
| 2 | ROH-RBP-muTTR | 0.4 | 1.6 | 0 | ROH partitions between free and RBP-bound states. RBP does not bind to muTTR. |

BMEC, brain microvascular endothelial-like cells; muTTR, mutant I84A transthyretin; RBP, retinol-binding protein; ROH, retinol; TTR, transthyretin.

*Distribution of ROH between unbound (free) and protein-bound states was calculated by assuming equilibrium and utilizing the known or measured $K_D$ for binding of RBP to ROH and for binding of TTR to ROH-RBP.

†Delivery mode indicates whether ROH was supplied to the cell culture medium in the absence of binding protein (free) or pre-complexed with RBP, muRBP (L63R/L64S), RBP-TTR, or RBP-muTTR (I84A). Total RBP and TTR concentrations were 2 µM and 4 µM, respectively.

In this report, we demonstrate the feasibility of using human iPSC-derived BMEC monolayers cultured on plasticware or permeable Transwells in conjunction with recombinant human RBP and TTR as an in vitro platform for studying retinoid uptake by, and transport across, the BBB. We have recently reported a robust method for producing and purifying recombinant human RBP in both apo- and holo-forms (*Est and Murphy, 2020*), as well as recombinant human wild-type (*Liu et al., 2009*) and mutant TTR (*Mangrolia et al., 2016*). To support the validity of this in vitro BBB model for retinoid transport studies, we confirmed expression of STRA6, LRAT, and CRBP1 protein in our iPSC model. We then investigated ROH accumulation and permeation across iPSC-derived BMECs when delivered by the two primary physiological sources: RBP and RBP-TTR complex. To further explore the utility of this experimental model, we compared accumulation and permeation of free lipid ROH to that of RBP-bound ROH. We also mutated RBP (L63R/L64S) and TTR (I84A) to modify their binding properties, providing a means to examine the role of each protein individually in ROH accumulation and permeation. Finally, we probed BMEC response to ROH uptake as a function of delivery mechanism by quantifying RNA expression levels for STRA6, LRAT, and CRBP1. Our results establish the utility of this platform for obtaining greater mechanistic insight into retinol trafficking across the BBB.

## Results

### iPSC-derived BMECs express BBB relevant phenotypes and key retinoid-related proteins

We verified that induced pluripotent stem cell-derived (iPSC) brain microvascular endothelial-like cells (BMECs) display BBB-relevant markers using commercially available antibodies (*Table 1*; Table 2), consistent with prior work (*Lippmann et al., 2014*; *Lippmann et al., 2012*). Specifically, iPSC-derived BMECs express (*Figure 2A*) endothelial cell marker PECAM-1; tight junction markers CLDN5, OCLN, and ZO-1; and GLUT1, a glucose transporter highly enriched at the BBB in vivo. Additionally, these cells express efflux transporters, including BCRP and MRP1, as expected (*Figure 2B*).

We next tested whether BMECs express key transporters and enzymes involved in retinoid metabolism and/or transport using commercially available polyclonal antibodies . Critically for our purpose, differentiated BMECs express STRA6 (*Figure 2B*), corroborating transcriptional evidence (*Lippmann*

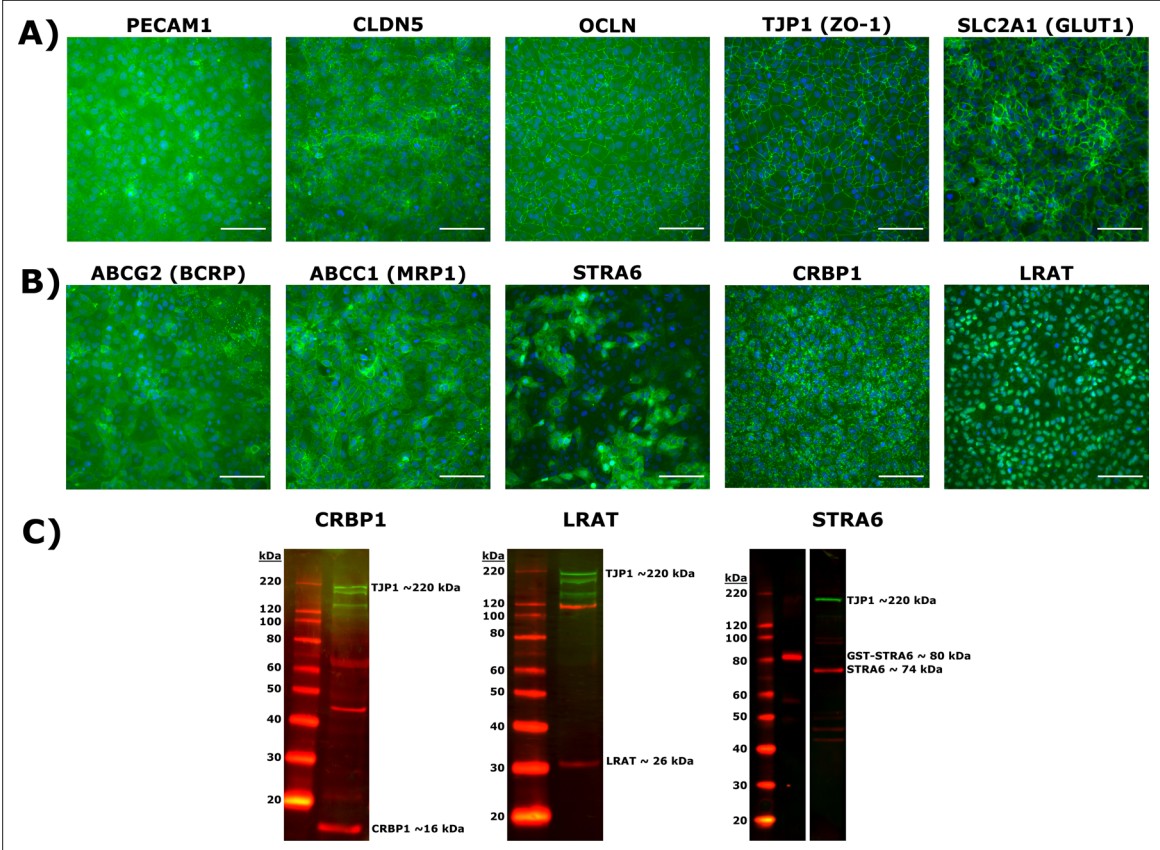

**Figure 2.** Induced pluripotent stem cells (iPSC)-derived brain microvascular endothelial-like cells (BMEC) marker and retinoid-related protein validation by immunocytochemistry and western blot. (**A**) Expression of endothelial cell marker PECAM-1; tight junction and associated proteins CLDN5, OCLN, and TJP1 (ZO-1); glucose transporter SLC2A1 (GLUT1). Proteins are labeled in green and nuclear stain in blue. Scale bar equals 100 μm. (**B**) Expression of efflux transporters ABCG2 (BCRP), ABCC1 (MRP1), and retinoid-related proteins STRA6, CRBP1, and LRAT. Proteins are labeled in green and nuclear stain in blue. Scale bars equal 100 μm. (**C**) Western blots of CRBP1 (red), LRAT (red), and STRA6 (red) confirming antibody specificity. An antibody against TJP1 (green) is used as a BMEC-specific loading control in each blot. Polyclonal STRA6 antibody was additionally validated against recombinant GST-tagged STRA6.

The online version of this article includes the following source data for figure 2:

**Source data 1.** Raw unedited gel CRBP1.

**Source data 2.** Raw unedited gel LRAT.

**Source data 3.** Raw unedited gel STRA6.

**Source data 4.** Annotated uncropped gels.

*et al., 2012*). Notably, STRA6 expression does not appear uniform. BMECs also express CRBP1, with a staining pattern consistent with CRBP1's cytosolic function (*Silvaroli et al., 2016*); similarly, BMECs express LRAT, with faint staining in the lipid bilayer and stronger staining near the cell nucleus consistent with LRAT's function in retinyl ester synthesis in lipid droplets (*O'Byrne and Blaner, 2013*; *Moise et al., 2007*). Immunocytochemistry results for STRA6, CRBP1, and LRAT were corroborated by western blotting using the same polyclonal antibodies as for the immunocytochemistry panel (Table 3, *Figure 2C*), with detection of bands at ~16 kDa (CRBP1), ~30 kDa (LRAT), and ~70 kDa (STRA6) as expected. In order to assuage concerns about commercially available antibodies against STRA6 (*Kawaguchi et al., 2015*), purified recombinant GST-tagged STRA6 protein was used as a positive control.

## Recombinant RBP and TTR are suitable replacements for serum sources

We previously expressed, purified, and characterized human wild-type TTR (*Liu et al., 2009*) and human wild-type retinol binding protein 4 (RBP) (*Est and Murphy, 2020*) in *Escherichia coli*. Recombinant

RBP binds retinol (ROH) with a dissociation constant $K_D$ = 100 ± 30 nM, indistinguishable from serum-derived human RBP (*Est and Murphy, 2020*). Recombinant RBP complexed with ROH (ROH-RBP) binds recombinant human TTR with a dissociation constant of ~250 nM, in close agreement with human serum-derived measurements (*Est and Murphy, 2020*).

## ROH accumulates in iPSC-derived BMEC monolayers from ROH-RBP or ROH-RBP-TTR complexes

BMEC monolayers cultured on 96-well plates were exposed to solutions of ROH-RBP or ROH-RBP-TTR prepared at biologically relevant concentrations: 2 µM ROH (typical range in vivo, 1–2 µM; *Jiménez-Jiménez et al., 1999*; *Zaman et al., 1992*), 2 µM RBP (in vivo range, 2–4 µM; *O'Byrne and Blaner, 2013*), and 4 µM TTR (in vivo range, 3–8 µM; *Hanson et al., 2018*). Solutions of ROH-RBP were prepared by overnight equilibration of free lipid ROH (1:20 ratio of $^3$H-ROH to unlabeled ROH) and ligand-free (apo) RBP. Solutions of ROH-RBP-TTR were prepared by overnight equilibration of ROH-RBP (holo) and TTR. After the desired incubation time for accumulation, cells were washed, lysed, and the tritium signal counted. ROH cellular accumulation, in µmoles of ROH per L cell volume, was calculated from DPM measurements using the manufacturer-supplied specific activity of $^3$H-ROH, the ratio of $^3$H-ROH to unlabeled ROH (1:20), and a cell volume of $1.37 \times 10^{-12}$ L/cell. Cell volume was estimated by multiplying the average area of a BMEC (402 µm$^2$) by its height (3.4 µm). BMEC area was estimated in two ways: directly from ICC image analysis and by counting the number of cells in a BMEC monolayer after singularization and dividing by the culture dish area. The two methods produced consistent estimates of cell area. BMEC height was estimated by analysis of Z-stack ICC images. We use the term 'ROH cellular accumulation' to represent the quantity of all added ROH that becomes cell-associated, recognizing that we did not determine whether any retinol was converted to oxidized metabolites or retinyl esters and we did not differentiate cell-associated radioactivity between internalized versus membrane-associated material after washing.

ROH cellular accumulation increased throughout the 2-hr experiment at a steady rate with either ROH-RBP or ROH-RBP-TTR (*Figure 3A*, *Figure 3—figure supplement 1*). TTR did not affect ROH cellular accumulation kinetics. After 2 hr, accumulated ROH cellular concentration was ~100 µM, or about 50-fold higher than the 2 µM ROH medium concentration. This indicates that the BMEC monolayer stores excess ROH and that ROH accumulates against a concentration gradient. High cellular ROH accumulation could be accounted for by a number of established mechanisms, including thermodynamically driven partitioning of the lipophilic ROH into lipid-rich cellular components, binding of ROH by intracellular proteins such as CRBP1, and/or storage of internalized ROH as retinyl esters (RE).

## BMEC monolayers are a useful in vitro BBB model system for measuring ROH permeation at physiologically relevant conditions

iPSC-derived BMEC monolayers have been shown to be a good in vitro model for the permeability of essential nutrients and drugs across the BBB (*Lippmann et al., 2014*; *Lippmann et al., 2012*). Vitamin A permeability has not previously been measured with iPSC-derived BMEC monolayers, and in fact there is a very sparse literature on retinol transport across the BBB in any in vitro or in vivo model (*MacDonald et al., 1990*; *Franke et al., 1999*; *Pardridge et al., 1985*). BMECs were cultured on semi-permeable Transwell inserts that allow for sampling of the apical chamber ('blood') and basolateral chamber ('brain') (*Figure 3B*). Transendothelial electrical resistance (TEER) was used to confirm the integrity of the BMEC barrier. TEER measures the resistance of the monolayer to an electrical current and correlates with the size of molecules excluded from paracellular transport. For compounds the size of sucrose or retinol (342 Da or 286 Da, respectively), a minimum TEER of ~500 $\Omega \times$ cm$^2$ is considered sufficient to exclude paracellular transport (*Mantle et al., 2016*). TEER for iPSC-derived BMECs in our study was typically 3000 $\Omega \times$ cm$^2$ at the start of incubations and remained above 1000 $\Omega \times$ cm$^2$ upon conclusion.

ROH-RBP or ROH-RBP-TTR solutions (using $^3$H-ROH as a tracer) were charged to the apical chamber at concentrations identical to the accumulation assay, and samples from both apical and basolateral chambers were collected at several time points over the course of the 1 hr experiment. We describe accumulated $^3$H signal in the basolateral chamber as retinoid, recognizing that we were not able to collect enough material to confirm its chemical identity specifically as retinol. $^{14}$C-sucrose was added to all samples as a control for barrier integrity since sucrose does not cross the in vivo BBB

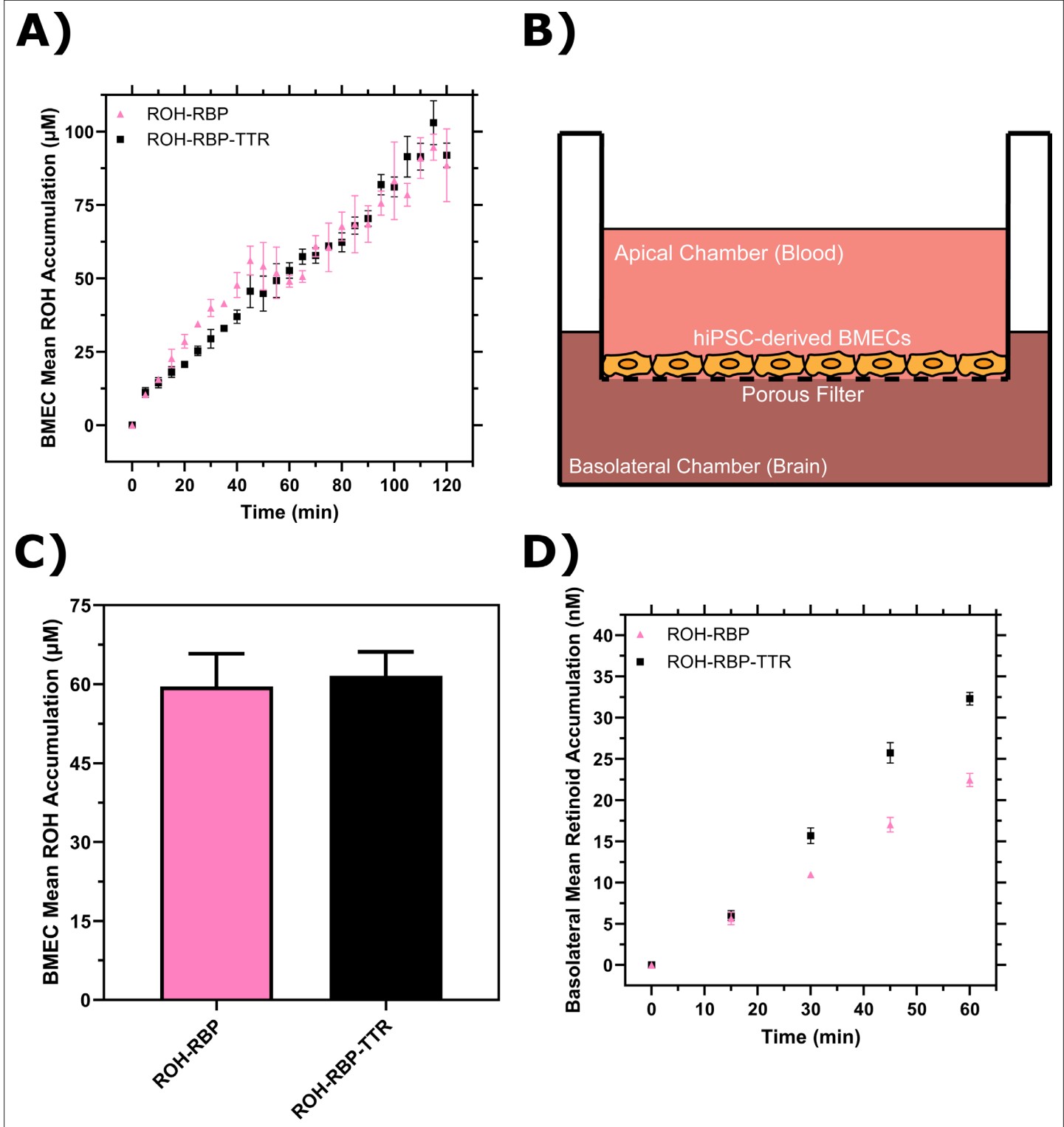

**Figure 3.** Brain microvascular endothelial-like cells (BMEC) retinol (ROH) uptake and permeation mediated by retinol-binding protein (RBP) or retinol-binding protein-transthyretin (RBP-TTR) complex. (**A**) Mean ROH cellular accumulation as a function of time. Measured DPM values were converted to accumulated concentrations using the specific activity of $^3$H-ROH, the $^3$H-ROH:unlabeled ROH ratio (1:20), and the average cell volume. Error bars represent the standard deviation of three biological replicates. Fluid concentrations are typical of human blood concentrations at 2 µM ROH, 2 µM RBP, and 4 µM TTR. (**B**) Schematic of the Transwell apparatus. The semi-permeable support allows for BMEC basolateral efflux. (**C**) Mean ROH cellular accumulation in BMEC lysate after 60 min, collected from cells in Transwells. Measured DPM values were converted to accumulated concentrations

*Figure 3 continued on next page*

*Figure 3 continued*

using the specific activity of [3]H-ROH, the [3]H-ROH:unlabeled ROH ratio (1:20), and the calculated cell volume. (**D**) Kinetics of retinoid accumulation in the basolateral chamber. Error bars represent the standard deviation of four biological replicates. Measured DPM values were converted to accumulated concentrations using the specific activity of [3]H-ROH, the [3]H-ROH:unlabeled ROH ratio (1:20), and the volume of the basolateral chamber medium. Apical concentrations were 2 µM ROH, 2 µM RBP, and 4 µM TTR. No RBP or TTR was added to the basolateral chamber.

The online version of this article includes the following source data and figure supplement(s) for figure 3:

**Source data 1.** Brain microvascular endothelial-like cells (BMEC) retinol (ROH) uptake and permeation mediated by retinol-binding protein (RBP) or retinol-binding protein-transthyretin (RBP-TTR) complex.

**Figure supplement 1.** Individual brain microvascular endothelial-like cells (BMEC) retinol (ROH) accumulation curves.

in large quantities (*Lippmann et al., 2012*). In these experiments, there is no RBP or TTR added to the basolateral chamber. At the conclusion of each experiment, BMEC monolayer lysate was collected to test for closure of the mass balance; ~97% of [3]H-ROH and at least 95% of [14]C-sucrose radioactivity was recovered (*Supplementary file 1a and b*). [3]H-ROH signal measured in lysates from Transwell experiments was used to calculate total cellular accumulation at the 1 hr time point (*Figure 3C*); cellular accumulation of ~60 µM is consistent with the monolayer results at 1 hr (*Figure 3A*) and again shows no statistical difference between ROH-RBP and ROH-RBP-TTR.

As shown in *Figure 3D*, there was a short lag period after which retinoid accumulation in the basolateral chamber increased linearly over the course of the 1-hr experiment. Interestingly, although ROH cellular accumulation was not affected by TTR in the apical chamber (*Figure 3A and C*), the basolateral retinoid accumulation was roughly 30% higher when TTR was present in the apical chamber. Basolateral retinoid accumulation after 1 hr was only a small fraction (1–1.5%) of the ROH concentration loaded in the apical chamber.

To quantify our data and confirm basolateral accumulation was due to permeability across the cellular monolayer rather than via paracellular leakage, we used *Equations 1 and 2* to calculate the apparent permeability ($Pe_{app}$) of the BMEC monolayer and Transwell semi-permeable insert combined for both sucrose and ROH (*Supplementary file 1c*). Sucrose $Pe_{app}$ ranged from $0.58 \pm 0.03$ to $0.69 \pm 0.04 \times 10^{-6}$ cm/s when mixed with ROH-RBP-TTR or ROH-RBP samples, respectively, in agreement with prior studies using these iPSC-derived BMECs ($Pe = 0.57 \times 10^{-6}$ cm/s) (*Lippmann et al., 2012*) and slightly lower (indicating a tighter barrier) than reported with primary porcine BMECs ($Pe = 1 \times 10^{-6}$ cm/s, TEER <1000 $\Omega \times$ cm$^2$) (*Franke et al., 1999*). These results confirm tight barrier integrity in this iPSC-derived model system. $Pe_{app}$ for ROH was $4.8 \pm 0.2 \times 10^{-6}$ cm/s when supplied by ROH-RBP and $6.6 \pm 0.3 \times \times 10^{-6}$ cm/s when supplied by ROH-RBP-TTR. These permeabilities are an order of magnitude higher than those for sucrose and in the same range as that of glucose ($Pe = 3.7 \times 10^{-6}$ cm/s; *Lippmann et al., 2012*), a critical nutrient for the brain. This analysis indicates that ROH is indeed transported transcellularly, and that inclusion of TTR increases the ROH permeation rate by ~30%. To our knowledge, this is the first report of permeability measurements for RBP-bound ROH across a BBB-like monolayer, as well as the first reported indicator that TTR may play a role in increasing permeation of ROH across the BBB.

## Free ROH cellular accumulation is bulk fluid concentration-dependent

In the presence of RBP and/or TTR, ROH partitions between free and protein-bound states. Although ROH circulating with RBP is thought to be the predominant mode of vitamin A delivery to cells and tissues, there is evidence that free ROH readily partitions into cell membranes (*Noy and Xu, 1990b*; *Noy and Xu, 1990a*; *Fex and Johannesson, 1990*; *Fex and Johannesson, 1988*). The estimated concentrations of free and protein-bound ROH at our experimental conditions were calculated from the known equilibrium dissociation constants and total concentrations of ROH, RBP, and TTR (*Table 1*).

For RBP, about 20% of ROH is free (0.4 µM free versus 1.6 µM bound to RBP), while when both RBP and TTR are present, only about 7% (0.14 µM) of ROH is estimated to be free. The relative contribution of free ROH to overall cellular accumulation and permeation across barriers is not known and has typically not been accounted for in other studies of RBP-mediated ROH delivery to cells (*Kawaguchi et al., 2007*; *Kawaguchi et al., 2012*; *Kawaguchi et al., 2013*; *Kawaguchi et al., 2011*; *Berry et al., 2012a*; *Kawaguchi et al., 2015*; *Kawaguchi and Sun, 2010*; *Kawaguchi et al., 2008*), although there is evidence cells respond differently to free ROH compared to ROH-RBP (*Zhong et al., 2014*).

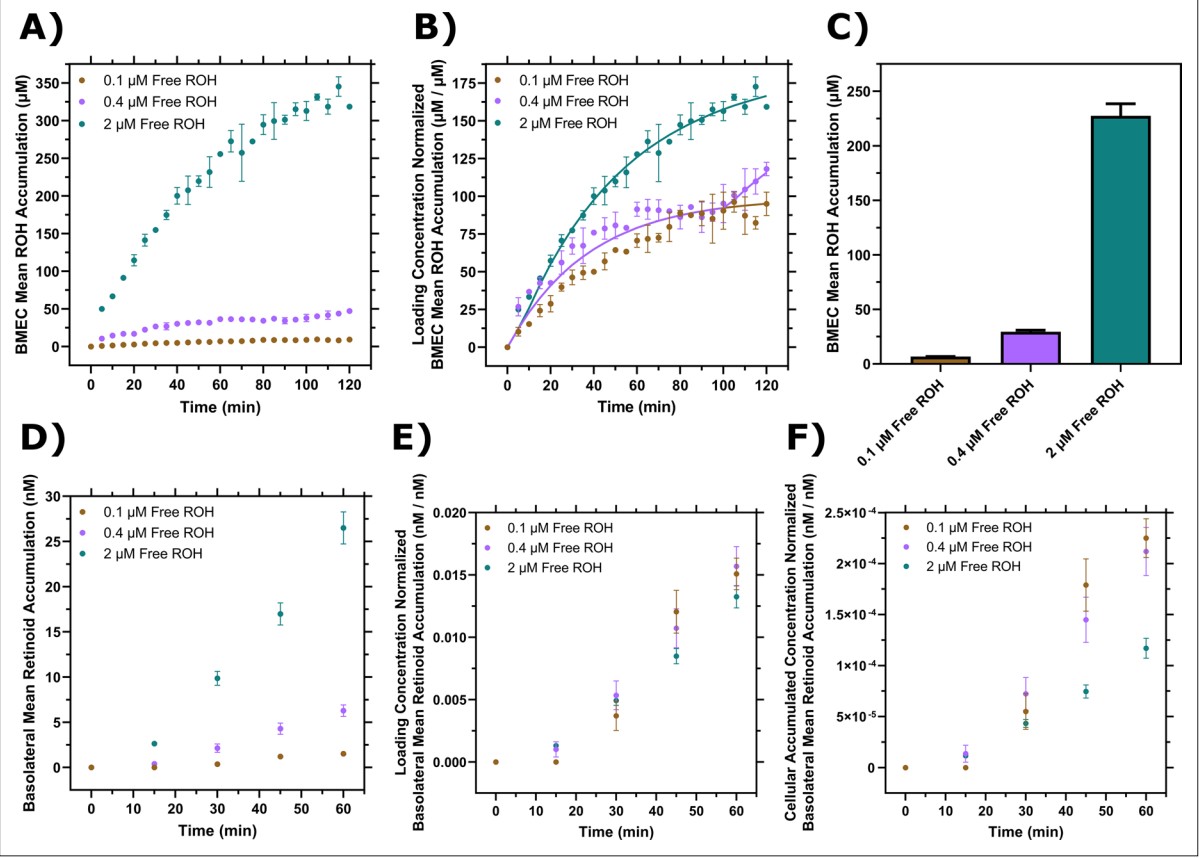

**Figure 4.** Brain microvascular endothelial-like cells (BMEC) retinol (ROH) uptake and permeation as a function of free ROH concentration. (**A**) Mean ROH cellular accumulation as a function of time and free ROH concentration. Measured DPM values were converted to accumulated concentrations using the specific activity of $^3$H-ROH, the $^3$H-ROH:unlabeled ROH ratio (1:20), and the average cell volume. Error bars represent the standard deviation of three biological replicates. Fluid concentrations were 0.1, 0.4, or 2 µM ROH. No retinol-binding protein (RBP) or transthyretin (TTR) was present in the medium. (**B**) Mean ROH accumulation from panel (**A**) normalized by the ROH concentration in the medium. Data are fit by a simple partitioning model with a secondary uptake mechanism *Equation 2* that triggers upon accumulation exceeding a fitted threshold value of ~36 µM. (**C**) Mean ROH cellular accumulation in BMEC lysate after 60 min, collected from cells in Transwells. Measured DPM values were converted to accumulated concentrations using the specific activity of $^3$H-ROH, the $^3$H-ROH:unlabeled ROH ratio (1:20), and the calculated cell volume. No RBP or TTR was added to either chamber. (**D**) Kinetics of retinoid accumulation in the basolateral chamber. Error bars represent the standard deviation of four biological replicates. Measured DPM values were converted to accumulated concentrations using the specific activity of $^3$H-ROH, the $^3$H-ROH:unlabeled ROH ratio (1:20), and the volume of the basolateral chamber medium. No RBP or TTR is added to either chamber. (**E**) Retinoid basolateral chamber accumulation normalized by the apical chamber ROH concentration. (**F**) Retinoid basolateral chamber accumulation normalized by the accumulated cellular ROH concentration at 60 min.

The online version of this article includes the following source data and figure supplement(s) for figure 4:

**Source data 1.** Brain microvascular endothelial-like cells (BMEC) retinol (ROH) uptake and permeation as a function of free ROH concentration.

**Figure supplement 1.** Mean retinol (ROH) accumulation partitioning model.

We used our experimental system to measure the free ROH concentration-dependent cellular accumulation and compare the data to cellular accumulation data from protein-bound ROH. BMEC monolayers were exposed to ROH at three concentrations: 2 µM (physiological), 0.4 µM (to approximate the free ROH concentration in ROH-RBP solutions), and 0.1 µM (to approximate the free ROH concentration in ROH-RBP-TTR solutions). Cell-associated ROH increased over the 2-hr time course of the experiment, with total accumulation a strong function of fluid-phase ROH concentration (*Figure 4A*). The accumulated cellular ROH concentration after 2 hr was nearly two orders of magnitude higher than the fluid-phase ROH concentration, consistent with the data observed for protein-bound ROH (*Figure 3A*). To examine concentration-dependent patterns of ROH accumulation kinetics, cellular concentrations were normalized by the ROH concentration initially loaded in the medium (*Figure 4B*), as the bulk fluid concentrations remained relatively stable over the course of the

experiment (*Supplementary file 1d*). This analysis demonstrates that the kinetic pattern is distinctly different at each ROH fluid-phase concentration. Briefly, at 0.1 μM ROH in the fluid phase, cellular concentration reached ~9 μM (or a cell:fluid ratio of ~90 μM/μM) by approximately 90 min, beyond which it remained stable. At 0.4 μM fluid-phase ROH, cellular concentration reached an apparent plateau of ~36 μM (cell:fluid ratio of ~90 μM/μM) at approximately 60 min, but then started to increase further at approximately 90 min. In contrast, at 2 μM fluid-phase ROH, cellular concentration increased continuously over the 2-hr experiment, reaching ~320 μM (~160 μM/μM cell/fluid concentration ratio).

We evaluated whether these data could be described by a simple kinetic model, where we assume the bulk concentration remains constant (*Supplementary file 1e*). The strong concentration dependence suggested a partitioning model (akin to solvent:solvent partitioning) as a better descriptor than a receptor-ligand binding model. If $c_f$ = bulk fluid concentration (μM), $c_{cell}$ = accumulated cellular concentration (μM) at any time $t$, $k_1$ = first-order rate constant (min$^{-1}$), $K_p$ = partition coefficient (μM cell/μM fluid), then a simple model is

$$\frac{c_{cell}}{c_f} = K_p \left[ 1 - exp\left(-k_1 t\right) \right] \tag{1}$$

However, given the apparent intermediate plateau for the 0.4 μM free ROH sample, as well as the observation that the ratio $c_{cell}/c_f$ increases with increasing fluid-phase concentration, we hypothesized that a secondary uptake mechanism is triggered after a lag time, $t_{lag}$, that corresponds to crossing an intracellular ROH threshold, $c_{cell}$*:

$$\frac{c_{cell}}{c_f} = K_p \left[ 1 - exp\left(-k_1 t\right) \right] + K_p^* \left[ 1 - exp\left(-k_1^* \left(t - t_{lag}\right)\right) \right] \tag{2}$$

where

$$t_{lag} = \frac{-ln\left[ 1 - \frac{c_{cell}^*}{K_p c_f} \right]}{k_1}$$

We fit the data by *Equation 1* (*Figure 4—figure supplement 1*) or *Equation 2* (*Figure 4*) using least-squares regression. Our analysis indicates that including the secondary uptake mechanism provides a significantly better description of the experimental data (*Figure 4—figure supplement 1* and *Supplementary file 1e*), providing support for the hypothesis of a biphasic response at higher fluid ROH concentrations. A possible explanation is that loading of CRBP1 with ROH to its capacity (estimated from our modeling to be $c_{cell}$* ~ 36 μM) triggers initiation of a secondary storage mechanism, such as retinyl ester synthesis, to handle additional ROH cellular accumulation. Indeed, biphasic ROH uptake from RBP in Sertoli cells has been reported; the accumulated ROH at the plateau in that study corresponded to the intracellular CRBP1 concentration (*Shingleton et al., 1989*).

## Free ROH permeates across the BBB-like barrier with kinetics proportional to apical concentration and not cellular concentration

Permeation of protein-free ROH at 0.1 μM, 0.4 μM, or 2 μM in the apical chamber was measured using the Transwell setup. To ensure barrier integrity, TEER and $^{14}$C-sucrose permeability were monitored as described previously. At the conclusion of each experiment, BMEC monolayer lysate was collected to test for closure of the mass balance; at least 90% of $^3$H-ROH and at least 96% of $^{14}$C-sucrose radioactivity was recovered (*Supplementary file 1a and b*). We first confirmed that cellular accumulation in this setup was consistent with that in the monolayer by measuring radioactivity in the lysate at the end of the permeation experiment (*Figure 4*). As shown in *Figure 4D*, basolateral retinoid accumulation rate increased with increasing apical ROH concentration. We calculated $Pe_{app}$ of ROH to range from 3.8 ± 0.4 to 7.7 ± 0.5 × 10$^{-6}$ cm/s as the apical ROH concentration increased from 0.1 to 2 μM. These values indicate that ROH is about 10-fold more permeable than sucrose (0.43 ± 0.03 to 0.58 ± 0.03 × 10$^{-6}$ cm/s) and are in good agreement with data reported for free ROH (supplied at ~10$^{-8}$ M) in primary porcine BMECs ($Pe$ = 4.1 ± 0.7 × 10$^{-6}$ cm/s) (*Franke et al., 1999*). The permeabilities are similar to those we calculated for ROH-RBP or ROH-RBP-TTR, demonstrating that RBP is not required for ROH transit across the BBB-like barrier. It is important to note that $Pe_{app}$ should be independent of the apical ROH concentration for a single permeation mechanism.

The observation that $Pe_{app}$ increases with increasing fluid-phase ROH concentration supports the hypothesis that higher fluid-phase ROH concentrations trigger secondary transport mechanisms in BMECs or that free ROH is processed intracellularly into different retinoid forms, such as retinyl esters.

To understand the concentration dependence of ROH permeation, we normalized the basolateral ROH concentration by the initial apical ROH concentration or by the accumulated cellular ROH concentration (*Figure 4E and F*). The data collapse to a single curve in *Figure 4E* but not in *Figure 4F*, demonstrating that transport of retinoid across the BMEC monolayer to the basolateral chamber is more tightly coupled to the apical chamber ROH concentration rather than to the accumulated concentration of ROH within the cell barrier.

## RBP and TTR markedly reduce ROH cellular accumulation compared to protein-free ROH, but permeability across the BBB-like barrier is similar for protein-bound and protein-free ROH

We asked whether cellular ROH accumulation kinetics differed between free ROH and protein-bound ROH when supplied at the same initial total ROH concentration. Accumulation kinetics for 2 µM free ROH (*Figure 4A*) are compared to accumulation kinetics for 2 µM ROH-RBP (*Figure 3A*) in *Figure 5A*. Since the calculated free ROH concentration in the 2 µM ROH-RBP sample at these conditions is ~0.4 µM (*Table 1*), data at 0.4 µM ROH (*Figure 4A*) are also included for comparison. ROH cellular accumulation levels from 2 µM ROH-RBP are substantially lower than at 2 µM free ROH, but slightly exceed that of 0.4 µM free ROH alone. For samples containing TTR (*Figure 5B*), the free ROH concentration is estimated to be ~0.14 µM (*Table 1*). ROH cellular accumulation levels from 2 µM ROH-RBP-TTR (*Figure 3A*) are markedly lower than those at 2 µM free ROH, but significantly exceed that at 0.1 µM free ROH (*Figure 4A*).

Cellular uptake of ROH in protein-bound samples could be achieved through two paths operating in parallel: directly from free ROH and/or by delivery from ROH bound to RBP or RBP-TTR. We wondered whether we could estimate the relative importance of these two paths by comparing cellular accumulation data from ROH alone versus ROH complexed to RBP or RBP-TTR (*Figure 5C*). Specifically, at each time point we normalized the free ROH cellular accumulation by the total ROH cellular accumulation measured for the protein-bound samples (0.4 µM free ROH normalized by ROH-RBP for RBP alone samples and 0.1 µM free ROH normalized by ROH-RBP-TTR for TTR-containing samples; see *Table 1*). For ROH-RBP, essentially 100% of the initial observed ROH cellular uptake could be accounted for by the contribution of the 0.4 µM free ROH present in the mixture. This percentage drops to 50% after 2 hr as ROH cellular accumulation from ROH-RBP samples grows more quickly than for 0.4 µm free ROH alone. This analysis is consistent with the hypothesis that free ROH partitions rapidly into the cell, but then that pathway's contribution diminishes over time as tightly controlled influx/efflux via the ROH-RBP route becomes more dominant. In contrast, ROH cellular accumulation from the 0.1 µM free ROH in ROH-RBP-TTR samples never exceeds ~20% of the total uptake. Thus, although the kinetics of cellular accumulation are virtually identical between ROH-RBP and ROH-RPB-TTR (*Figure 3A*), the *relative* contributions of free versus protein-delivered ROH appear to be quite different (*Figure 5C*). This illustrates the importance of careful accounting for all bound and free forms of ROH in mechanistic investigations.

Although *cellular* accumulation was much higher for 2 µM free ROH than when bound to RBP or RBP-TTR, permeation into the *basolateral* chamber was similar (but not identical) for all three cases, again highlighting that permeation across the BMEC barrier is correlated more strongly with the total apical ('blood') ROH concentration and not with the free apical ROH concentration or the cellular ROH accumulated concentration (*Figure 5D*). However, basolateral permeation was not entirely a function of total apical ROH concentration, but also depended on whether ROH was presented with RBP alone or in complex with TTR; permeation with RBP alone was lower than for an equivalent total concentration of free ROH, whereas permeation with RBP in complex with TTR was higher than for an equivalent concentration of free ROH. This observation suggests that RBP and TTR might influence ROH cellular trafficking even though the proteins are not themselves internalized, and that the proteins may have subtle differences in the observed effect.

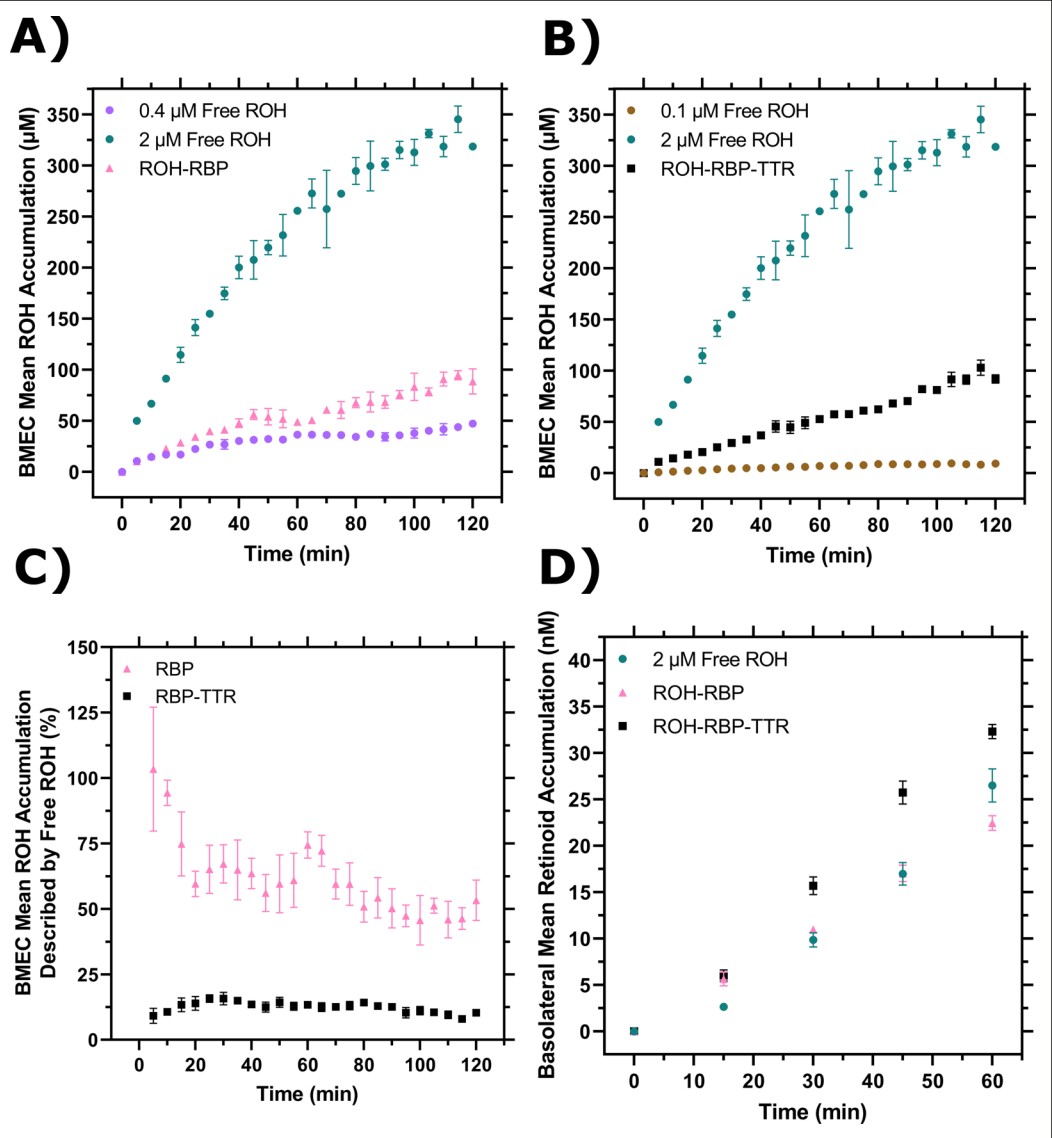

**Figure 5.** Comparison of cellular retinol (ROH) accumulation and permeation with or without retinol-binding protein (RBP) or transthyretin (TTR). (**A**) Cellular accumulation from ROH-RBP, replotted from *Figure 3A*, compared to 0.4 μM or 2 μM ROH, replotted from *Figure 4A*. The ROH-RBP solution is an equilibrated mixture of 2 μM ROH and 2 μM RBP, with a calculated protein-free (unbound) ROH concentration of 0.4 μM. (**B**) Cellular accumulation from ROH-RBP-TTR, replotted from *Figure 3A*, compared to 0.1 μM or 2 μM ROH, replotted from *Figure 4A*. The ROH-RBP-TTR solution is an equilibrated mixture of 2 μM ROH, 2 μM RBP, and 4 μM TTR, with a calculated protein-free (unbound) ROH concentration of 0.14 μM. (**C**) Percentage of cellular ROH accumulation from ROH-RBP or ROH-RBP-TTR that could be potentially described by the contribution of free ROH present in the protein samples (0.4 μM free ROH for ROH-RBP and 0.1 μM for ROH-RBP-TTR). (**D**) Comparison of basolateral retinoid permeation for free ROH, ROH-RBP, and ROH-RBP-TTR samples each containing 2 μM total ROH per sample.

## RBP and TTR mutants reveal novel insights into mechanisms of ROH cellular accumulation in, and permeation across, BMEC monolayers

To further explore the role RBP and TTR play individually in ROH cellular accumulation and permeation, we generated mutants that abrogate wild-type binding interactions. L63R/L64S mutations in RBP (muRBP) alter a loop region at the entrance of the β-barrel of RBP (*Figure 6A*, adapted from *Perduca et al., 2018*), and this loop region is critical in mediating the binding interaction to TTR. Furthermore, muRBP has been previously shown unable to bind to a protein expressed on placental membranes that participates in retinol transport (*Sivaprasadarao and Findlay, 1994*); it is suspected,

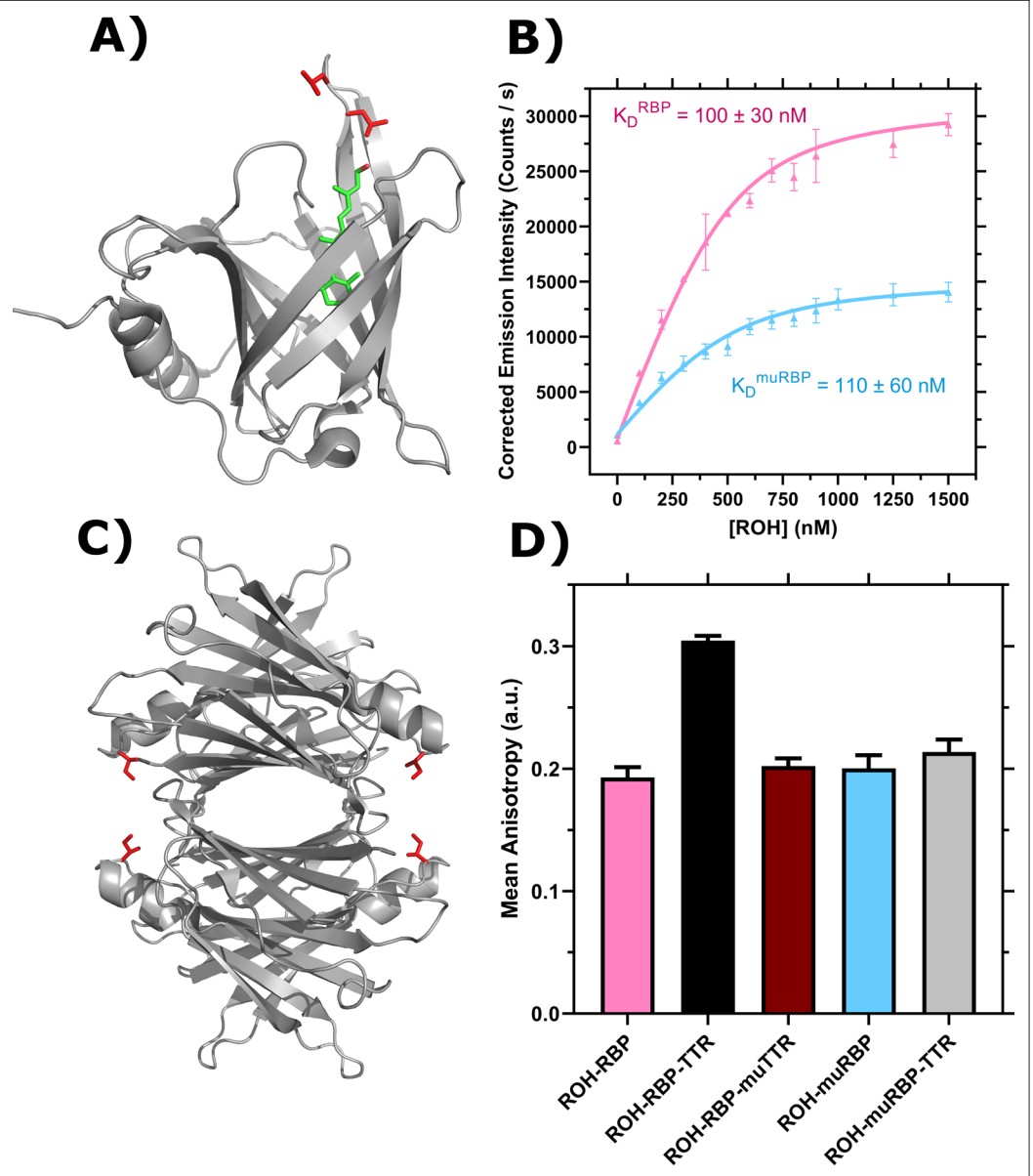

**Figure 6.** Characterization of retinol-binding protein (RBP), transthyretin (TTR), and mutants. (**A**) PDB entry 5NU7 (***Perduca et al., 2018***). Ribbon diagram displays RBP with bound retinol (ROH) shown in green and L63 and L64 in red. (**B**) ROH binding to RBP (pink, adapted from ***Est and Murphy, 2020***) and muRBP (blue) monitored by emission of ROH at 460 nm as an acceptor in resonance energy transfer from donor RBP tryptophan. Data are fit by nonlinear regression as described (***Est and Murphy, 2020***). Error bars represent the standard deviation of three independent replicates. (**C**) PDB entry 5CN3 (***Yee et al., 2016***). Ribbon diagram displays the TTR tetramer. I84 for each TTR monomer is shown in red. (**D**) Representative binding of 1 µM ROH-RBP or 1 µM ROH-muRBP to 4 µM TTR or 4 µM muTTR as measured by fluorescence anisotropy using the polarized emission of ROH at 460 nm. Error bars represent the standard deviation of three technical replicates.

The online version of this article includes the following source data and figure supplement(s) for figure 6:

**Source data 1.** Characterization of retinol-binding protein (RBP), transthyretin (TTR), and mutants.

**Figure supplement 1.** Normalized chromatographic analysis of wild-type retinol-binding protein (RBP) and L63R/L64S mutant RBP.

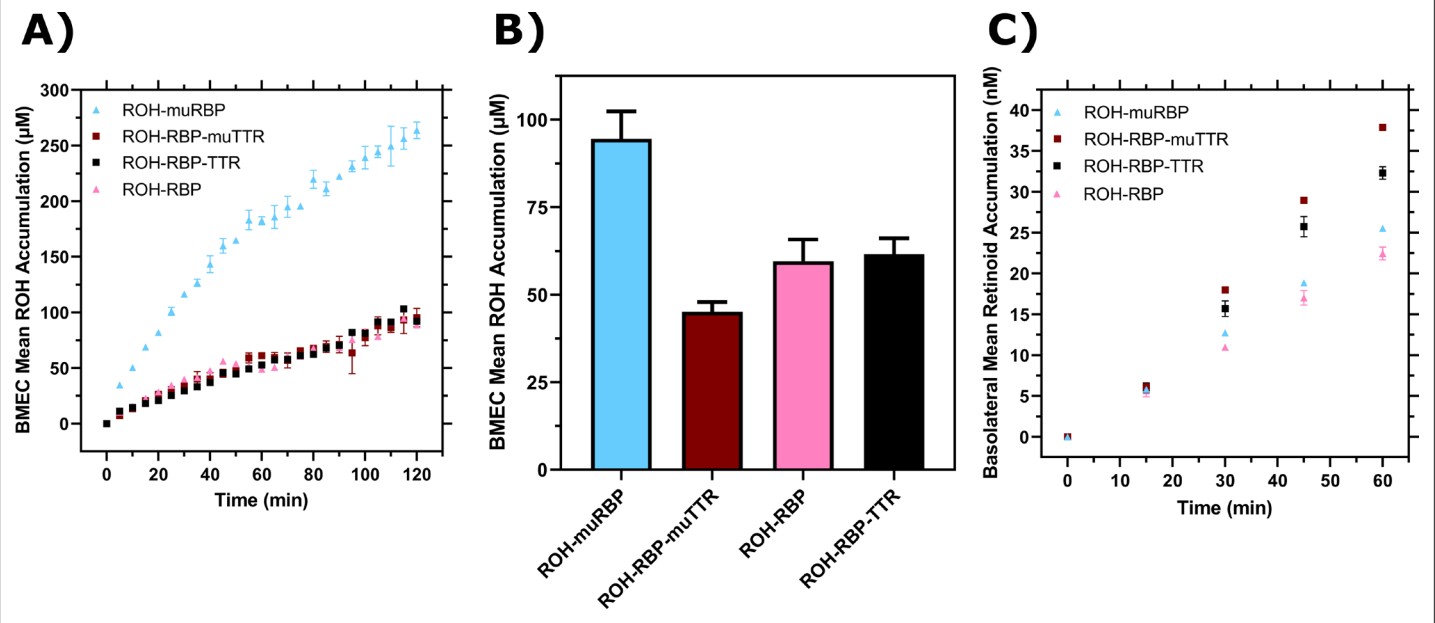

**Figure 7.** Comparison of retinol-binding protein (RBP) and transthyretin (TTR) to muRBP and muTTR. (**A**) Mean retinol (ROH) cellular accumulation as a function of time and concentration. Error bars represent the standard deviation of three biological replicates. The medium contains 2 μM ROH equilibrated with 2 μM RBP, 2 μM muRBP, 2 μM RBP +4 μM TTR, or 2 μM RBP +4 μM muTTR. (**B**) Mean ROH cellular accumulation in brain microvascular endothelial-like cells (BMEC) lysate after 60 min, collected from cells in Transwells. Apical concentrations are the same as listed in panel (**A**). (**C**) Kinetics of retinoid permeation into the basolateral chamber comparing ROH-RBP, ROH-muRBP, ROH-RBP-TTR, and ROH-RBP-muTTR.

The online version of this article includes the following source data for figure 7:

**Source data 1.** Brain microvascular endothelial-like cells (BMEC) retinol (ROH) uptake and permeation mediated by retinol-binding protein (RBP) or retinol-binding protein-transthyretin (RBP-TTR) complex or their mutants.

but has not been confirmed, that this protein is STRA6. We produced and purified muRBP, and purity was confirmed using analytical size-exclusion chromatography (aSEC) and DEAE anion-exchange chromatography. RBP and muRBP elute in one primary peak on both aSEC and DEAE columns. On aSEC, muRBP elutes earlier than RBP, suggesting the L63R/L64S mutation causes an increase in the apparent volume of the protein. However, refolded ROH-muRBP displays an $A_{330/280}$ ratio of ~1.0 in the presence of excess ROH, indicating normal 1:1 binding stoichiometry (data not shown). Furthermore, ROH binds to muRBP with $K_D$ = 110 ± 60 nM, comparable to RBP (**Figure 6B**, **Figure 6—figure supplement 1**). Interestingly, muRBP fluorescence at saturation is only ~50% the magnitude of ROH-RBP, which we theorize is due to changes in the β-barrel local environment that alter the efficiency of resonance energy transfer; this hypothesis would be consistent with the apparent larger volume seen by aSEC. To confirm loss of TTR-binding affinity, we utilized a fluorescence anisotropy assay that demonstrates ROH-muRBP does not bind TTR (**Figure 6D**).

To examine the role of TTR binding to RBP on retinol uptake and trafficking, we produced an I84A mutant TTR (muTTR) with reduced affinity for RBP (**Monaco, 2000**; **Du et al., 2012**; **Figure 6C**, adapted from **Yee et al., 2016**). Fluorescence anisotropy confirmed that recombinant muTTR did not bind measurably to ROH-RBP (**Figure 6D**).

First, we measured ROH cellular accumulation in BMEC monolayers using muRBP or muTTR and compared the results against RBP or TTR. We found negligible differences in ROH cellular accumulation between ROH-RBP and ROH-RBP-muTTR (**Figure 7A**); this result was expected since muTTR does not bind ROH-RBP.

Surprisingly, accumulation from ROH-muRBP is markedly *higher* than that of ROH-RBP; indeed, accumulation approaches that of free ROH at 2 μM. This was an unexpected result given that muRBP binds ROH with similar affinity as RBP, but may be consistent with the hypothesis that muRBP cannot participate in ROH efflux from intracellular ROH stores, and thereby drives a higher net ROH accumulation.

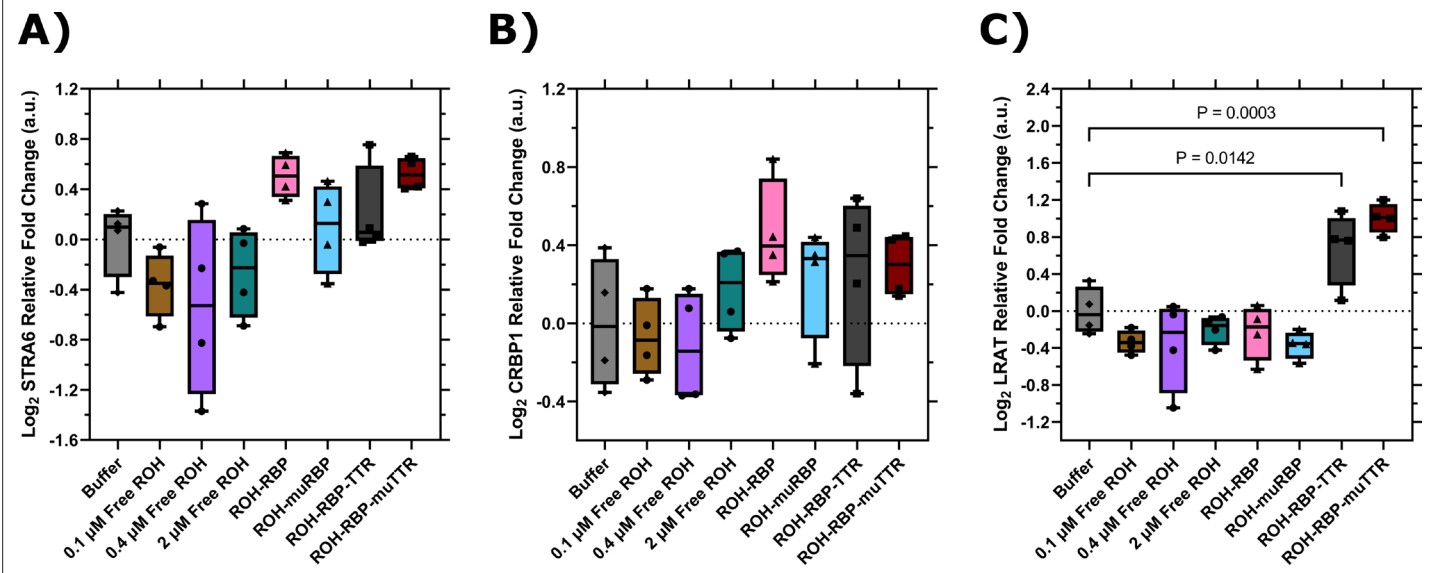

**Figure 8.** RT-qPCR data for brain microvascular endothelial-like cells (BMECs) treated with different retinol (ROH) modalities for 2 hr. Expression values are normalized to the housekeeping gene, ACTB, and quantified relative to BMECs treated with HBSS alone. ΔΔCq data are presented in box-and-whisker format after log$_2$ transformation with the values for each biological replicate displayed individually (N = 4). Statistical analyses were performed in Prism on log$_2$ transformed ΔΔCq values via one-way ANOVA followed by Dunnett's test using a confidence interval of 95%. (**A**) STRA6; (**B**) CRBP1; and (**C**) LRAT.

The online version of this article includes the following source data for figure 8:

**Source data 1.** RT-qPCR data for brain microvascular endothelial-like cells (BMECs) treated with different retinol (ROH) modalities for 2 hr.

We next measured ROH permeability across the BMEC monolayer in the Transwell configuration using muRBP or muTTR with RBP. We confirmed consistency of the monolayer plate and Transwell experiments, showing again that cellular ROH accumulation was significantly higher with muRBP than with RBP (*Figure 7B*). Although *cellular* accumulation was much higher for ROH bound to muRBP than compared to RBP, *basolateral* retinoid permeation was virtually identical (*Figure 7C*). This result is consistent with our other observations that basolateral permeation is primarily correlated with apical and not cellular ROH concentration.

Because the I84A TTR mutant does not bind to RBP, we anticipated that ROH-RBP-muTTR would demonstrate permeability similar to ROH-RBP alone. However, basolateral permeability was higher in samples containing TTR or muTTR when compared to ROH-RBP alone (*Figure 7C*). This surprising result indicates that it is the presence of TTR, and not specifically the binding of TTR to RBP, that is responsible for the higher basolateral permeation of ROH.

## TTR and muTTR upregulate expression of LRAT

It is known that influx of ROH delivered by ROH-RBP through STRA6 triggers an intracellular signaling cascade (*Berry et al., 2011*; *Chen et al., 2012*). STRA6-mediated transport of ROH is coupled to intracellular CRBP1 (*Berry et al., 2012b*; *Kawaguchi et al., 2011*) concentrations, while LRAT plays an important role in storing excess intracellular ROH by enzymatically converting it to retinyl esters. We used our in vitro system to explore whether ROH uptake may trigger transcriptional changes in these vitamin A-related proteins when treated with ROH and its binding partners. Specifically, we used RT-qPCR to measure changes in expression of STRA6, CRBP1, and LRAT after treatment of BMECs with the ROH preparations listed in *Table 1*. Primer sequences used are detailed in Table 4.

Neither STRA6 nor CRBP1 expression was affected to a statistically significant degree by addition of ROH, RBP, or TTR (*Figure 8A and B*). Nor was there a significant change in LRAT expression when BMECs were exposed to free ROH, ROH-RBP, or ROH-muRBP (*Figure 8C*). Samples containing either ROH-RBP-TTR or ROH-RBP-muTTR, however, show statistically significant upregulation of LRAT (*Figure 8C*). Since TTR binds to RBP but muTTR does not, LRAT upregulation is independent of TTR binding to RBP and is decoupled from the delivery of ROH through STRA6 via RBP. Since LRAT plays

a role in managing intracellular ROH inventory, this result could explain why the presence of TTR, and not its binding to RBP, increases ROH permeability across the BMEC monolayer (*Figure 7C*). Although more detailed investigation is needed, our data are indicative of a heretofore unknown function for TTR in regulating ROH delivery across barriers, independent of its well-known role as a carrier for RBP.

## Discussion

Despite the importance of retinoids to brain health, very little is known about the mechanisms by which retinoids enter the specialized cells forming the BBB or how retinoids transit the BBB. Using human iPSC-derived BMECs, as well as recombinant human RBP and TTR, we constructed an in vitro platform that provides a practical means for controlled experimentation and quantitative interrogation of retinol uptake by, and delivery across, the BBB. iPSC-derived BMECs express markers indicative of brain endothelial specificity, including PECAM1, Claudin-5, Occludin, ZO-1, and Glut1. BCRP and MRP1 expression provide evidence of efflux transporter activity, a major function of the in vivo BBB. In this work, we show that BMECs express STRA6 as well as two other vitamin-A relevant proteins: LRAT and CRBP1, both of which are known to couple with STRA6 for ROH uptake. Recombinant RBP and TTR were produced that are suitable replacements for human plasma-derived materials while providing a means to probe the roles of these proteins individually in ROH trafficking and transport via specific mutations. Using this in vitro system, we measure for the first time the rate of ROH transport across a human BBB-like barrier and show that ROH permeability is similar to that of an essential brain nutrient, glucose.

Our results clearly show that cells take up free ROH in a concentration-dependent manner, and that ROH binding proteins are not required for cellular accumulation (*Figure 4A*). After 2 hr, cell-associated ROH concentrations exceed that in the fluid phase by two orders of magnitude. Since transport of free ROH into the cell is maintained against a strong concentration gradient, this result indicates that iPSC-derived BMECS store ROH primarily in a bound or associated form. This is similar to another blood barrier, the retinal pigment epithelium (RPE), where 92% of total intracellular ROH is stored as RE and 8% as ROH-CRBP1 complex (*Napoli, 2016*).

The kinetic pattern of BMEC ROH uptake varied with ROH concentration (*Figure 4B*). The biphasic behavior of ROH uptake in Sertoli cells is similar to what we observed in iPSC-BMECs at intermediate concentrations of ROH; the plateau in that study occurred at an intracellular ROH concentration corresponding to the CRBP1 concentration (*Shingleton et al., 1989*). We propose that, at 0.1 µM fluid-phase ROH, BMECs maintain an adequate sink for intracellular ROH, possibly through binding to CRBP1. At higher ROH concentrations, ROH accumulates intracellularly beyond the CRBP1 binding capacity, and a secondary storage system is recruited into action. A simple kinetic model was developed that is consistent with this proposed mechanism and provides an estimate of the 'triggering' cellular ROH concentration as ~36 µM. The secondary storage system may proceed through LRAT, which normally esterifies ROH bound to CRBP1 and is regulated by the ratio of bound to unbound CRBP1; however, in periods of excess free ROH, LRAT can act on free ROH directly (*Ong et al., 1988*; *Herr and Ong, 1992*). Further study is needed to validate this hypothesis.

Not only can free vitamin A be taken up by BMECs, but it also can be transported across the barrier without requiring RBP or TTR (*Figure 4D*, see *Figure 1B*). Notably, basolateral ('brain') accumulation was directly proportional to the apical ('blood') ROH concentration, and *not* to the cellular concentration. This suggests that cells at the BBB store excess ROH and carefully regulate retinoid delivery to the 'brain' in response to blood concentrations and/or that protein-delivered ROH has greater ability to access trans-cellular trafficking pathways than ROH taken up as free lipid.

RBP carries out two key functions: it sequesters ROH in the blood in order to maintain free ROH concentrations at sub-toxic levels, and it delivers ROH to cells via STRA6. We tested whether ROH accumulation in, or retinoid transport across, BMECs was enhanced by RBP. Notably, complexation of ROH with RBP *slows* cellular accumulation considerably relative to what occurs in the absence of RBP at an equivalent ROH concentration (compare ROH-RBP versus 2 µM free ROH, *Figure 5A*). Interpretation of this result requires consideration of the fact that there is ~0.4 µM free ROH in the ROH-RBP preparation, and a better comparison therefore should account for the contribution of the 0.4 µM free ROH present in the ROH-RBP preparation. We hypothesize that cellular uptake of ROH occurs by two parallel paths, from free ROH and from ROH-RBP. A direct comparison of the kinetics (*Figure 5C*) leads to our hypothesis that free ROH rapidly partitions into the cell, whereas the RBP-mediated

cellular accumulation is slower; after 2 hr, however, about half of the total ROH cellular concentration may be accounted for by each of the pathways.

Experiments with the L63R/L64S mutant RBP provide further insight into the role of RBP in regulating ROH trafficking. Cellular uptake of ROH is significantly faster with muRBP compared to RBP, and accumulation reached levels approaching that of 2 μM free ROH. This was an unexpected result since muRBP binds ROH with equal affinity to RBP (*Figure 6B*). As illustrated in *Figure 1B*, ROH can cross cellular membranes through multiple mechanisms: by passage of free ROH through the lipid bilayer, by passage through cell-surface proteins, or by STRA6-mediated release of ROH from ROH-RBP. Importantly, RBP-mediated transport through STRA6 is bidirectional, with both influx and efflux depending on availability of intracellular apo-CRBP1 and extracellular apo-RBP, respectively (*Kawaguchi et al., 2012*). Furthermore, ROH efflux requires binding of apo-RBP to STRA6 (*Kawaguchi et al., 2012*; *Kawaguchi et al., 2011*), a mechanism available to RBP but presumably not muRBP (*Sivaprasadarao and Findlay, 1994*). We postulate that cellular accumulation is therefore the net sum of three steps: uptake of free fluid-phase ROH, uptake of ROH via delivery from holo-RBP, and efflux of cellular ROH via uptake by extracellular apo-RBP. The substantially higher cellular accumulation for muRBP compared to RBP provides support for the hypothesis that *efflux* of ROH is a critical regulatory component for BMEC intracellular ROH levels and that this pathway is inoperable with muRBP. Additional support for this hypothesis would require direct characterization of the binding of RBP and muRBP to our iPSC-derived BMECs, a subject for future research. Regardless, these data suggest that the most important role of RBP during times of retinol abundance is to mediate efflux of excess intracellular ROH, thereby maintaining intracellular ROH at sufficient, but not excessive, concentrations.

Despite the large differences in ROH cellular accumulation with 2 μM free ROH versus the same quantity incubated with RBP or muRBP, the ROH permeation kinetics were very similar (*Figure 7C*). This result suggests that BMEC monolayers regulate transport of ROH across the barrier by 'sensing' the apical ('blood') ROH concentration. A plausible mechanism for achieving 'sensing' is via the intracellular ratio of ROH-bound CRBP1 and ligand-free apo-CRBP1. Apo-CRBP1 promotes RE hydrolysis (*Herr and Ong, 1992*; *Boerman and Napoli, 1991*) and influx of ROH through STRA6 (*Kawaguchi et al., 2012*), while ROH-CRBP1 promotes RE synthesis via LRAT (*Ong et al., 1988*) and efflux of ROH through STRA6 (*Kawaguchi et al., 2012*). Efflux of ROH through STRA6 requires availability of ligand-free RBP (apo-RBP) in the blood. Therefore, the ratio of intracellular ROH-CRBP1 to apo-CRBP1 is directly coupled via STRA6 to blood ROH through the ratio of extracellular ROH-RBP to apo-RBP.

TTR's primary role in ROH transport is generally believed to be in binding to RBP and preventing loss of the small protein through the kidney (*Monaco, 2000*). ROH is more buried in the ROH-RBP-TTR complex compared to ROH-RBP (*Monaco, 2000*), and cross-linking studies show evidence of an ROH-RBP-STRA6 complex but not of an ROH-RBP-TTR-STRA6 complex (*Berry et al., 2012a*); taken together, these observations suggested that dissociation of ROH-RBP from TTR is required for RBP-mediated transfer of ROH to STRA6. Indeed, we observed that ROH cellular accumulation with ROH-RBP-TTR was indistinguishable from ROH-RBP (*Figure 3A*), which at first glance would indicate that TTR does not play a direct role in ROH uptake into the BMEC monolayer (see *Figure 1Biii*). However, there are significant differences in the ROH distribution between ROH-RBP and ROH-RBP-TTR delivery modalities that may suggest a more complicated scenario: first, the free ROH concentration is ~4× lower when both RBP and TTR are present (~0.14 μM, see *Table 1*) than compared to RBP alone (0.4 μM, see *Table 1*); second, the concentration of ROH-RBP differs by an order of magnitude between RBP and TTR containing samples (0.18 μM, see *Table 1*) and the RBP alone samples (1.6 μM, see *Table 1*). The comparative analysis shown in *Figure 5C* indicates that direct uptake of free ROH in ROH-RBP-TTR mixtures contributes much less to the overall cellular accumulation than when compared to ROH-RBP mixtures alone. Since there is no evidence of direct binding of ROH-RBP-TTR to STRA6, we postulate that ROH-RBP in ROH-RBP-TTR mixtures must play a much larger role in ROH cellular influx than when ROH-RBP is presented alone. This may have significant physiologic ramifications; STRA6-mediated JAK/STAT signaling is activated specifically by ROH *influx* via ROH-RBP (see *Figure 1Bii*; *Berry et al., 2012b*); furthermore, this signaling cascade requires ROH to be delivered directly from RBP, as neither RBP nor ROH alone induce the cascade (*Berry et al., 2011*).

When TTR was added to the apical chamber, we observed higher permeation into the basolateral chamber, and a higher $Pe_{app}$ than that of free ROH or ROH-RBP at the same total apical ROH concentration (*Figure 7C*). Surprisingly, this effect was seen with both TTR and muTTR, despite the fact that

muTTR does not bind to ROH-RBP. This result indicates that it is the presence of TTR, and not the binding of TTR to RBP, that is responsible for the higher permeation of ROH. To gain further insight, we looked for changes in expression of STRA6, CRBP1, or LRAT. No statistically significant changes in STRA6 or CRBP1 expression were detected after two hours of incubation. With LRAT, however, we saw a statistically significant increase in expression in BMECs exposed to either TTR or muTTR, but not to ROH or ROH-RBP alone. This is a novel finding that suggests TTR plays an important role in regulating retinol trafficking and transport across barriers. Moreover, this regulatory activity does not require TTR to be complexed to RBP. If TTR signaling directly increases expression of LRAT, concomitant enhanced production of RE could serve as the substrate for basolateral efflux, the mechanistic basis of which is wholly unknown. Further studies are required to tease out this unexpected role of TTR in retinol processing.

Taken together, our results demonstrate the utility of our in vitro BBB model constructed from iPSC-derived BMECs and recombinant wild-type and mutant RBP and TTR for studies of retinol trafficking across the BBB. Several novel findings include (1) accumulation of ROH in the cells of the BBB is a strong function of the delivery mode (free or protein-bound), while permeation across the BBB is mostly independent of delivery mode, (2) retinol permeation rates across the BBB are similar to that of glucose, another essential brain nutrient, (3) efflux of ROH through STRA6 to apo-RBP in the serum may be an underappreciated route for controlling intracellular accumulation in times of retinol abundance, and (4) TTR upregulates LRAT expression and influences ROH transport to the brain using mechanisms that are independent of its RBP-binding role. We highlight the importance of using wild-type and mutant RBP and TTR, as well as careful accounting for the distribution of ROH between free and protein-bound states, in any mechanistic investigation of retinol trafficking, permeability, or STRA6-mediated signaling.

By leveraging the renewable, scalable, and genetically manipulable features of human iPSC-derived BMECs, the roles of key cellular proteins involved in retinoid accumulation, metabolism, and permeation across the BBB could be directly explored. Recent advances in CRISPR-mediated perturbation methods, such as knockout (CRISPR$_{ko}$), interference (CRISPR$_i$), or activation (CRISPR$_a$), can be used to map the roles of STRA6, LRAT, and/or CRBP1 in retinoid processing at the BBB. Results reported here lay the groundwork for more detailed investigations into mechanisms of retinol transport into the brain, which are expected to yield greater insights into retinol's role in supporting brain health and provide novel approaches for treatment of retinol dysfunction in neurodegenerative disease.

# Materials and methods
## iPSC differentiation to BMECs

IMR90-4 iPSCs (iPSC) were provided by WiCell (Madison, WI). Certificate of analysis is available here. Authentication analyses include karyotyping and identity by STR. Mycoplasma testing by PCR confirmed negative. IMR90-4 iPSCs were cultured on Matrigel-coated 6-well plates and supplemented daily with E8 medium (Stem Cell Technologies) as described (*Stebbins et al., 2016*). iPSCs were passaged in clumps at ~70% confluency every 3–5 d by dissociation with Versene (Life Technologies) at typical ratios between 1:6 and 1:12. To initiate differentiation into BMECs, iPSCs at ~70% confluency were dissociated and singularized by treatment with Accutase (Life Technologies) for 7–10 min and then diluted into fresh E8 media (1:4 *v/v*). Cells were counted on a hemocytometer, and then centrifuged at $200 \times g$ for 5 min. The supernatant was aspirated and the pellet resuspended in fresh E8 medium supplemented with 10 µM ROCK inhibitor (Tocris R&D Systems). Resuspended cells were seeded at a density between 7500 and 12,500 cells/cm$^2$ on fresh Matrigel-coated plates (day 3). ROCK-supplemented E8 media was aspirated and replaced with fresh E8 media (without ROCK inhibitor) 24 hr later to promote iPSC colony formation. Cells were subsequently expanded for 48 hr with daily E8 media replacement until reaching the optimal density of 30,000 cells/cm$^2$ as described previously (*Wilson et al., 2015*). Unconditioned medium (UM: 50 mL knock-out serum replacement, 2.5 mL non-essential amino acids, 1.25 mL GlutaMAX [all from Life Technologies], and 1.75 µL of β-mercaptoethanol [Sigma] diluted into Dulbecco's Modified Eagle Medium/Nutrient Mixture F-12 [DMEM] to 250 mL and vacuum filtered into 0.2 µm PES filter-top bottles) was prepared and stored at 4°C for up to 2 wk. On day 0, E8 medium was replaced with UM replenished daily for 6 d. On day 6, UM media was replaced with fresh EC +/+ media for 48 hr without replacement. EC +/- medium was prepared

as follows: 5 mL B-27 Supplement (50×), serum free (Thermo Fisher) was diluted into human endothelial serum-free medium (Life Technologies) to 250 mL, and vacuum filtered into 0.2 µm PES filter-top bottles. EC +/- medium was stored at 4°C for up to 2 wk. EC +/+ media was prepared fresh daily by supplementation of EC +/- media with 10 µM all-*trans* retinoic acid (Sigma, diluted in DMSO) and FGF2 (WiCell/Waisman Biomanufacturing) diluted 1:5000. On day 7, plasticware (Corning) was coated with a 4:1:5 *v/v/v* collagen IV (Sigma)/fibronectin (Sigma)/sterile water stock solution. For Transwell-Clear permeable inserts (0.4 µm pore size), the concentration of collagen and fibronectin, respectively, was 400 µg/mL and 100 µg/mL. Each Transwell filter was coated with 200 µL of 4:1:5 solution. For all other plasticware, 4:1:5 solution was further diluted fivefold in sterile water. On day 8, cells were singularized with Accutase for 30–45 min, resuspended in EC +/+ medium, and plated onto the 4:1:5 collagen IV/fibronectin-coated plasticware prepared fresh the day before at a density of $1 \times 10^6$ cells/cm$^2$ for 1.12 cm$^2$ Transwell-Clear permeable inserts, at a density of 250,000 cells/cm$^2$ on 6/12-well tissue culture polystyrene plates, or at a density of $1 \times 10^6$ cells/cm$^2$ for 24-/48-/96-well tissue culture polystyrene plates. On day 9, EC +/+ medium was replaced with EC +/- medium without retinoic acid to promote barrier tightening. At least one Transwell was seeded per differentiation to monitor TEER as a measure of BMEC quality. TEER was measured every 24 hr after subculture on day 8 to confirm barrier tightness. Resistance was recorded using an EVOM ohmmeter with STX2 electrodes (World Precision Instruments Sarasotae, FL). Maximum TEER values for iPSC-derived BMECs prepared by this protocol are typically >2000 Ω × cm$^2$, calculated by multiplying the resistance readings by 1.12 cm$^2$ to account for the Transwell surface area. Maximum TEER was typically observed on day 10, on which all subsequent experiments were performed.

## Immunocytochemical analysis of BMECs

BMECs were singularized on day 8 and seeded on 96-well plates coated with 4:1:5 solution. On day 10, cells were rinsed once with Dulbecco's phosphate-buffered saline (DPBS) and fixed with either 100% methanol or 4% paraformaldehyde for 10–15 min at room temperature. After fixation, the fixing agent was aspirated and the cells were washed three times in immediate succession with DPBS. After washing, cells were incubated for 60 min at room temperature in blocking buffer (10 % *v/v* goat serum in DPBS) before overnight incubation at 4°C with primary antibodies diluted in blocking buffer (*Table 2*).

After primary incubation, cells were washed three times at 5 min each with DPBS before incubation for 1 hr at room temperature in the dark with secondary antibodies in blocking buffer (goat anti-rabbit IgG (H+L) Cross-Adsorbed Secondary Antibody conjugated to Alexa Fluor 488 (Thermo Fisher, 1:200 dilution, Cat# A-11008) or goat anti-mouse IgG1 Cross-Adsorbed Secondary Antibody conjugated to Alexa Fluor 488 (Thermo Fisher, 1:200 dilution, Cat# A-21121)). After secondary antibody incubation, cells were immediately stained for 15 min at room temperature in the dark with Hoechst nuclear count stain (Thermo Fisher) diluted 1:5000 in DPBS. Cells were washed three times and visualized in fresh

**Table 2.** Antibodies and staining information for immunocytochemistry.

| Antibody target | Manufacturer | Catalog number | Clonality | Host | Dilution | Fixative |
|---|---|---|---|---|---|---|
| PECAM1 | Thermo Fisher | RB-10333-P1 | Polyclonal | Rabbit | 1:25 | MeOH |
| CLDN5 | Thermo Fisher | 35-2500 | Monoclonal | Mouse | 1:200 | MeOH |
| OCLN | Thermo Fisher | 33-1500 | Monoclonal | Mouse | 1:50 | MeOH |
| TJP1 (ZO-1) | Thermo Fisher | 33-9100 | Monoclonal | Mouse | 1:200 | MeOH |
| SLC2A1 (GLUT1) | Thermo Fisher | MA1-37783 | Monoclonal | Mouse | 1:500 | MeOH |
| ABCG2 (BCRP) | MilliporeSigma | MAB4155 | Monoclonal | Mouse | 1:50 | 4% PFA |
| ABCC1 (MRP1) | MilliporeSigma | MAB4100 | Monoclonal | Mouse | 1:25 | MeOH |
| STRA6 | Abnova | H00064220-D01P | Polyclonal | Rabbit | 1:200 | 4% PFA |
| CRBP1 | Thermo Fisher | PA5-28713 | Polyclonal | Rabbit | 1:500 | MeOH |
| LRAT | Thermo Fisher | PA5-38556 | Polyclonal | Rabbit | 1:250 | MeOH |

**Table 3.** Antibodies and staining information for western blot.

| Antibody target | Manufacturer | Catalog number | Clonality | Host | Dilution |
|---|---|---|---|---|---|
| LRAT | Thermo Fisher | PA5-38556 | Polyclonal | Rabbit | 1:500 |
| CRBP1 | Thermo Fisher | PA5-28713 | Polyclonal | Rabbit | 1:1000 |
| STRA6 | Abnova | H00064220-D01P | Polyclonal | Rabbit | 1:1000 |
| TJP1 (ZO-1) | Thermo Fisher | 33-9100 | Monoclonal | Mouse | 1:1000 |
| TJP1 (ZO-1) | Thermo Fisher | 40-2200 | Polyclonal | Rabbit | 1:1000 |

DPBS on a Nikon Ti2 epifluorescence microscope with a ×20 objective. Images were analyzed with FIJI software.

## Western blot analysis for LRAT, CRBP1, and STRA6 expression

BMECs were singularized on day 8 and seeded on 6-well plates coated with 4:1:5 solution. On day 10, BMECs were rinsed once with DPBS (Thermo Fisher) and lysed for 15 min at 4°C using ice-cold radioimmunoprecipitation assay buffer (Rockland) supplemented with protease inhibitor cocktail (Pierce) according to the manufacturer's recommendation. Lysate, including cell membrane debris, was collected via scraping and centrifuged at max speed for 10 min at 4°C. Supernatant was collected and the protein concentration quantified by bicinchoinic acid assay according to the manufacturer's protocols (Pierce). Lysate was prepared for SDS-PAGE by mixing with 4× NuPAGE LDS Sample Buffer and NuPAGE Sample Reducing Agent. Samples were boiled at 70°C for 10 min and loaded into 4–12% Bis-Tris SDS-PAGE gels (Thermo Fisher) at 10–20 μg protein/well. SDS-PAGE was run at 200 V using either NuPAGE MES SDS Running Buffer to resolve LRAT and CRBP1 or NuPAGE MOPS SDS Running Buffer to resolve STRA6. All running buffers were supplemented with NuPAGE Antioxidant according to the manufacturer's protocols. Samples were transferred at 30 V in NuPAGE Transfer Buffer to 0.45 micron polyvinylidene difluoride (PVDF) membranes (Amersham) for 1 hr. After transfer, PVDF membranes were washed three times at 5 min each in Tris-buffered saline with 0.1% Tween 20 (TBST). Membranes were blocked for 1 hr at room temperature in blocking buffer (5 % *w/v* non-fat dry milk dissolved in TBST) before overnight incubation at 4°C with primary antibodies diluted in blocking buffer (*Table 3*).

After primary incubation, membranes were washed three times at 5 min each in TBST before incubation for 1 hr at room temperature in the dark with goat anti-rabbit IgG IRDye 680RD (Li-Cor, 1:5000 dilution, Cat# 926-68071) and donkey anti-mouse IgG IRDye 800CW (Li-Cor, 1:5000 dilution, Cat# 926-32212) secondary antibodies in blocking buffer. Membranes were washed three times for 5 min each in TBST and imaged using a Li-Cor Odyssey Imager.

## Preparation of retinol, RBP, and TTR

All-*trans* retinol (Sigma) and alpha-tocopherol (Sigma), which served as an antioxidant stabilizer, were dissolved in ethanol in equimolar concentrations and stored at –80°C. Concentrations of retinol (ROH) and alpha-tocopherol were determined by absorption using molar extinction coefficients of 52,480 $M^{-1}$ $cm^{-1}$ at 325 nm and 3260 $M^{-1}$ $cm^{-1}$ at 292 nm, for ROH and alpha-tocopherol, respectively. [15-3H(N)]-ROH (American Radiolabeled Chemicals), with an activity of 30 Ci/mmol, was supplied in an ethanol solution at 1 mCi/mL stabilized by alpha-tocopherol at a concentration of 1 mg/mL and stored at –20°C.

Recombinant human retinol-binding protein IV (RBP) and mutant L63R/L64S retinol-binding protein IV (muRBP) were produced in *E. coli* as inclusion bodies using the pTWIN system as described (*Est and Murphy, 2020*). Briefly, bacteria were lysed by sonication and the insoluble fraction collected by centrifugation. The inclusion body pellet was denatured and reduced in guanidine chloride supplemented with dithioerythritol (DTT), and then refolded in a cysteine/cystine oxidative buffer in the presence of excess all-*trans*-ROH. Refolded ROH-RBP or ROH-muRBP was purified on a chitin affinity column and released by intein-mediated self-cleavage. High-quality ROH-RBP and ROH-muRBP preparations were confirmed by $A_{330/280}$ ratios of ~1.0, indicating ROH binding at 1:1 stoichiometry, and these preparations were stripped of retinol by successive liquid–liquid extractions via anhydrous

diethyl ether. Stripped RBP was sparged with nitrogen to remove trace diethyl ether, and subsequently concentrated and buffer exchanged into phosphate-buffered saline (PBS) via centrifugal filtration. Concentrations were quantified via absorption using an extinction coefficient of 40,400 $M^{-1}$ $cm^{-1}$ at 280 nm. ROH-RBP and ROH-muRBP used for experiments were prepared as equimolar solutions 24–48 hr prior to use. Critically, confirmation of ROH binding was determined by absorption at 330 nm. The equilibrium dissociation constants for ROH to RBP and ROH to muRBP were both measured using fluorescence spectroscopy as described (*Est and Murphy, 2020*). [3]H-ROH-RBP and [3]H-ROH-muRBP were prepared identically using [15-3H(N)]-retinol.

Recombinant human TTR and mutant I84A transthyretin (muTTR) were prepared using the pTWIN system as described (*Liu et al., 2009*). Briefly, protein was recovered from inclusion bodies by sonication in 8 M urea buffer, centrifugation to remove any insoluble material and rapid dilution of supernatant into Tris buffer to a final urea concentration of 4 M. The protein solution was applied to a chitin affinity column and allowed to refold on column. TTR was eluted by intein-mediated self-cleavage, and purified protein was dialyzed into PBS and concentrated via centrifugal filtration prior to storage at 4°C. Concentrations were determined via absorption using an extinction coefficient of 77,600 $M^{-1}$ $cm^{-1}$ at 280 nm. ROH-RBP-TTR and ROH-RBP-muTTR samples used for experiments were prepared 24 hr prior to use by mixing TTR or muTTR with stock solutions of ROH-RBP. Confirmation of TTR binding to ROH-RBP was determined by fluorescence anisotropy, where binding of TTR increases ROH anisotropy (Ex/Em: 330/460). [3]H-ROH-RBP-TTR and [3]H-ROH-RBP-muTTR complex were prepared identically using [3]H-ROH-RBP stock solutions.

## Retinol cellular accumulation

BMEC monolayers cultured in 96-well plates were prepared as described and were used on day 10 for retinol accumulation assays, performed at 37°C on a shaker. BMECs from the same differentiation were seeded in parallel on Transwells to confirm quality via TEER measurements. Briefly, on day 10 of the differentiation, cell media was aspirated from all wells and replaced with solutions composed of HBSS (Thermo Fisher) and ROH, ROH-RBP, ROH-muRBP, ROH-RBP-TTR, or ROH-RBP-muTTR. Unlabeled ROH was mixed 1 hr prior to use with a [3]H tracer prepared identically at a target final concentration of 5% [3]H-ROH. The actual concentration was quantified by liquid scintillation and tracer percentage adjusted accordingly. For each experiment, one 96-well plate was subdivided into 25 triplicate groups and loaded with the various ROH preparations. Wells were aspirated serially in triplicate every 5 min over the course of 2 hr, and each well was considered a biologically distinct replicate. Immediately after aspiration, each well was washed twice with HBSS and allowed to dry. All wells were lysed simultaneously by addition of 100 µL of ice-cold radioimmunoprecipitation assay (RIPA) buffer per well and incubated at 4°C for 10 min. Following lysis, each 100 µL lysate/RIPA mixture was placed into an individual liquid scintillation counting vial and measured for [3]H DPM. Briefly, vials were diluted with 10 mL of UltimaGold (PerkinElmer), shaken vigorously, and counted immediately three times for 5 min each on a Tri-Carb 2900TR Liquid Scintillation Counter (PerkinElmer). The average DPM from the three technical replicate readings was utilized as the readout. Tritiated samples were counted using a preset region of 0–18.6 keV, while carbon-14 samples, where required, were counted simultaneously using a preset region of 0–156 keV. Self-normalization and calibration (SNC) was performed using external standards prior to each data run. DPM were converted to cellular concentrations by using the specific activity of the tritiated retinol and an assumed cellular volume calculated by multiplying the average area of a cell by its average height.

## Two-step RT-qPCR

BMECs were singularized on day 8 and seeded on 24-well plates coated with 4:1:5 solution. On day 10, BMECs were treated in parallel with the relevant ROH preparations for 2 hr. Each condition tested contained three biological replicates. Wells were aspirated after 2 hr and cells immediately incubated in Accutase for 30 min at 37°C to promote dissociation. After 30 min, cells were collected in microcentrifuge tubes and centrifuged at 200 × *g* for 5 min. The supernatant was aspirated and samples were stored at –80°C. RNA from each individual sample was harvested independently using the QIAGEN RNeasy Mini Kit (QIAGEN) according to the manufacturer's instructions. To prepare cell lysate, samples were suspended in RNeasy kit Lysis Buffer and β-mercaptoethanol (1:100 *v/v*) and homogenized using QIAshredder columns (QIAGEN). Samples loaded on spin columns were treated

**Table 4.** qPCR primer information.

| Gene | Expected product size (bp) | Oligo sequences<br>5′ – forward primer – 3′<br>5′ – reverse primer – 3′ | Final reaction concentration (nM) |
|---|---|---|---|
| LRAT | 149 | AGCCTGCTGTGGAACAACTG<br>GCCAATCCCAAGACTGCTGA | 100 |
| CRBP1 | 175 | AGATCGTGCAGGACGGTGA<br>CCCTTCTGCACACACTGGAG | 100 |
| STRA6 | 216 | ACACACAGGACCAACCTTCGAG<br>GAGCACAGGCATGAGCACCA | 200 |
| ACTB | 218 | CATCCGCAAAGACCTGTACG<br>CCTGCTTGCTGATCCACATC | 100 |

with DNase (QIAGEN) to digest genomic DNA prior to RNA purification. RNA was eluted in molecular biology grade water (Corning) and quantified by UV-vis absorbance using an Eppendorf BioSpectrometer. Sample quality was confirmed by a clean absorbance scan with an $A_{260/280}$ ratio of >1.8. Then, 200 ng of purified RNA was immediately reverse transcribed at 37°C for 1 hr in an S1000 Thermal Cycler (Bio-Rad) via the OmniScript RT Kit (QIAGEN) according to the manufacturer's instructions using RNase Out (Thermo Fisher) and Oligo(dT)$_{12-18}$ primers (Thermo Fisher). cDNA was used immediately or stored at 4°C for no more than 24 hr prior to qPCR.

qPCR samples were prepared using 2× PowerUP SYBR Green (Thermo Fisher), 10 ng of cDNA template prepared previously, and PCR primers diluted in molecular biology grade water as described in *Table 4*.

Samples were loaded onto qPCR-skirted plates (Agilent) and placed into an AriaMx Real-time PCR System (Agilent). Each run was initiated with a hot start at 95°C for 15 min, followed by 50 cycles consisting of a 15 s denaturation step at 95°C, followed by a 60 s amplification step at 60°C. Fluorescence was analyzed after each cycle. After 50 cycles, the PCR product was melted and melt curves were inspected after each run to ensure only one peak was observed, corresponding to the appropriate PCR product as determined by agarose gel electrophoresis. Each condition analyzed contained four biological replicates with three technical replicates for each gene. An arithmetic mean Cq value was calculated from the technical replicates for each gene, and this value for BMECs treated with HBSS alone was used as the reference to calculate ΔCq values for each biological replicate of each gene for each ROH delivery condition. ΔCq values were converted to normalized expression quantities using an assumed 100% efficiency and ACTB as a housekeeping gene. Statistical analyses were performed on log$_2$-transformed relative fold change quantities using one-way ANOVA followed by Dunnett's test with an assumed confidence interval of 95%. Log$_2$-transformed relative fold change data are presented graphically in box-and-whisker format.

## Retinol BBB permeability assays

BMEC monolayers cultured on collagen/fibronectin 12-well Transwells were prepared as described. On day 10 of the differentiation, TEER was optimal and BMECs were used for permeability assays. Then, 550 µL of ROH-containing solution was applied in triplicate to the apical chambers and 1500 µL of HBSS was added into each basolateral chamber. ROH preparations were spiked with $^{14}$C-sucrose (PerkinElmer) as a paracellular transport control. Samples collected from the apical and basolateral chambers over the course of the experiment totaled 10% of the bulk volume, and liquid removed from basolateral chambers was immediately replenished with the same volume of fresh HBSS. At $t$ = 0, samples were collected from both apical and basolateral chambers. Plates were then placed at 37°C on a shaker. Samples were collected from the basolateral chambers every 15 min. At $t$ = 60 min, an additional sample was also collected from the apical chamber. Once removed, each sample was placed into a scintillation vial for counting. TEER was measured after the final sample collection at 60 min to verify integrity of the monolayer. At the conclusion of the assay, solutions in both the apical and basolateral compartments were aspirated and 100 µL of ice-cold RIPA buffer was added directly to the BMEC monolayer to lyse the cells. After incubation at 4°C for 10 min, cell lysate was scraped and collected for scintillation counting. Concentrations of tritium in the apical and basolateral

chambers were calculated using the specific activity and well volume. Concentrations of tritium associated with the BMEC monolayers were calculated using the specific activity and the estimated volume of a single BMEC cell.

## Acknowledgements

IMR90-4 iPSCs were a generous gift from Professors Eric Shusta and Sean Palecek. Images used to calculate the area of a single BMEC were collected and provided by Benjamin Gastfriend. Schematics of ROH transport at the BBB were created with BioRender. We thank Professor Eric Shusta and Ben Gastfriend for helpful discussions on BMEC differentiation and development of permeability assays. Funding was provided by the Wisconsin Alzheimer's Disease Research Center's Pilot Funding Program and the Wisconsin Alzheimer's Disease Research Center Lightning REC Award. Additional support was provided by an NIH Biotechnology Training Grant to UW-Madison (NIH 5T32 GM008349). The content is solely the responsibility of the authors and does not necessarily represent the official views of the National Institutes of Health.

## Additional information

### Funding

| Funder | Grant reference number | Author |
| --- | --- | --- |
| Wisconsin Alzheimer's Disease Research Center | Pilot Funding Award and Lightning REC Award | Regina M Murphy |
| National Institutes of Health | Biotechnology Training Grant 5 T32 GM008349 | Regina M Murphy |

The funders had no role in study design, data collection and interpretation, or the decision to submit the work for publication.

### Author contributions

Chandler B Est, Conceptualization, Data curation, Formal analysis, Investigation, Methodology, Writing – original draft, Writing – review and editing; Regina M Murphy, Conceptualization, Formal analysis, Supervision, Funding acquisition, Project administration, Writing – review and editing

### Author ORCIDs

Chandler B Est ⓘ https://orcid.org/0000-0002-5588-9930
Regina M Murphy ⓘ https://orcid.org/0000-0002-6196-5450

Reviewer #1 (Public Review): https://doi.org/10.7554/eLife.87863.2.sa1
Reviewer #2 (Public Review): https://doi.org/10.7554/eLife.87863.2.sa2
Reviewer #3 (Public Review): https://doi.org/10.7554/eLife.87863.2.sa3
Author Response https://doi.org/10.7554/eLife.87863.2.sa4

## Additional files

### Supplementary files

• Supplementary file 1. This file contains supporting data on mass balances, permeability, accumulation and partitioning. (a) Mass balances for $^3$H-ROH. (b) Mass balances for $^{14}$C-sucrose. (c) Mean apparent permeability of $^3$H-ROH and $^{14}$C-sucrose. (d) Maximum percentage of fluid phase ROH accumulated by cells. (e) Partitioning model fits for BMEC free ROH accumulation.

### Data availability

Data generated or analyzed during this study are included in the manuscript and supporting files. Source Data files have been provided for Figures 2, 3, 4, 6, 7, 8.

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
