## [Editor Report · eLife assessment]

This **fundamental** work substantially advances our understanding of retinol transport through the blood–brain barrier. The evidence supporting the conclusions is **compelling**, with rigorous biochemical assays. In general, the work is of broad interest to cell biologists, biochemists, and neuroscientists.

---

## [Referee Report · Reviewer #1 (Public Review)]

This study provides a novel in vitro model for the study of retinol transport across the human BBB by pairing iPSC-derived BMECs with the use of recombinant vitamin A serum transport proteins, RBP and TTR. Key findings of the paper include (1) the observation that the delivery mode of retinol affects its intracellular accumulation at the BBB but not its permeation across the BBB, (2) further highlighting that intracellular concentrations of retinol are also ensured by its efflux via its receptor STRA6 and (3) a potential novel role for TTR in retinol transport by upregulating LRAT mRNA expression, independently of RBP. Notably, the model appears to be more accurate than ones previously used (primary porcine BMECs) to study retinol delivery at the BBB, and could be used to study the retinol dysregulation at the BBB in neurodegenerative diseases (e.g. by using iPSC lines from NDD patients), something that miss in the paper.

Indeed, the major disappointment of this work is the clinical relevance that was highlighted in the Introduction but was never really studied in the end. iPSC from patients could be added to the study.

As a general comment, the study is well done however the introduction and the discussion as a bit long and do not get to the point of the work easily. Even sometimes losing the reader in many details (necessary here?). Less abbreviations would be appreciated for general readers.

---

## [Referee Report · Reviewer #2 (Public Review)]

The manuscript by Est and Murphy tested the feasibility of using brain microvascular endothelial-like cells (BMECs) derived from induced pluripotent stem cells (iPSCs) as a model for studying retinoid uptake and transport across the blood-brain barrier (BBB). Establishing this experimental model is an important step towards obtaining greater mechanistic insight into the specificity of retinol trafficking between blood and retinoid-dependent tissues. The authors validated the iPSC-derived BMECs by detecting the expression of specific protein markers for BBB. They also demonstrated that BMECs form a tight barrier when cultured in a Transwell chamber, allowing for the quantification of permeability across the cells rather than through paracellular leakage. Finally, they confirmed the expression of the transporter (STRA6), binding protein (CRBP1), and enzyme (LRAT), which are key elements of the molecular machinery involved in the cellular uptake of circulating retinol. The carefully established model of the human BBB served as an experimental platform for the authors to investigate the uptake and transcellular transport of retinol. For this purpose, they compared the kinetics and efficiency of retinoid accumulation delivered to the cell as free retinol, retinol bound to serum retinol-binding protein (RBP), or retinol-RBP in complex with transthyretin (TTR), a physiological binding partner for retinol-loaded RBP.

Although the development and thorough characterization of the experimental model of the BBB have great value and meaningfully contribute to ongoing efforts to better understand the mechanisms of retinoid homeostasis, the premise and interpretation of cellular uptake appear controversial. In particular:

1. The authors assume that there is a significant fraction of free ROL, 20% for ROH/RBP and 7% for RBP/TTR complexes (summarized in Table 1). This implies that at the physiological concentration of ROH/RBP in the plasma of 2 uM, free ROL represents 0.4 uM. However, the concentration of free ROL is limited by its poor solubility in the aqueous phase, which is around 0.06 uM (Szuts EZ, 1991, Arch Biochem Biophys). Moreover, taking into account the large concentration of other potential nonspecific carriers for lipids, it is safe to assume that there is virtually no free ROH in the plasma. There is also an important physiological reason for the limited amount of free ROL. Its rapid and nonspecific partition into cells (also observed in this study) would work against the highly specific RBP/STRA6-dependent ROH uptake pathway, undermining its physiological function.

2. The advantage of the experimental system used in this report is that it allows for the assessment of the permeability across BMECs. Interestingly, the basolateral accumulation of ROH represented only a small fraction (1 - 1.5%) of the total ROH taken up by the cells. Moreover, the overall permeability was comparable regardless of the source of ROL added at the apical side. However, a question remains: would the outcome of the experiment be different if the basolateral chamber contained an ROH acceptor (retinol-binding proteins) rather than Hank's balanced salt solution, to which the partition of ROL is limited by its water solubility? In fact, the maximum concentration of ROH on the basolateral side did not exceed 40 nM (Fig 5D and 7C), which is roughly the maximum water solubility of ROH. Thus, this experimental design limits extrapolation of the data to in vivo conditions.

3. The authors claim that transthyretin (TTR) increases BMECs permeability when compared to ROH/RBP. However, the mechanistic explanation for this phenomenon remains unclear. Do the authors imply the presence of a putative TTR receptor whose signaling could affect the efflux of ROL at the basolateral side of BMECs? TTR is an ubiquitous plasma protein. The concentration of TTR is tightly regulated and maintained between 300 - 330 mg/L. Therefore, it is questionable how TTR can serve as a signaling molecule modulating retinoid homeostasis in the brain.

4. Although overexpression of LRAT in response to increased uptake of ROH is well-documented, the postulate that TTR stimulates the expression of LRAT in an RBP-independent manner is puzzling, for the reasons mentioned in point 3. Moreover, LRAT is a highly efficient enzyme that operates under physiological conditions with substrate concentrations below the Km value. The rate of esterification is primarily limited by the intracellular transport of ROH to the ER. Therefore, without kinetic studies, it is unclear whether an increased number of LRAT copies (x2) would have a significant effect on the rate of accumulation of retinyl esters (REs).

5. The conclusion that cellular uptake of ROH is biphasic appears to be correct. However, the proposed interpretation of the mechanistic principles of this phenomenon is oversimplified. It assumes that loading CRBP1 with ROL to its capacity triggers the synthesis of REs. However, the saturation of CRBP1 with ROH is not required for REs formation. In fact, studies on CRBP1-deficient mice indicate that this protein is not necessary for the efficient esterification of ROL but rather affects the intracellular turnover of retinoids. It is likely that with increasing concentration of ROH, the specific and controlled mechanism of intracellular retinoid transport becomes saturated, allowing for spontaneous diffusion-driven partitioning of retinoids within cells.

Additional technical issues that could affect the experimental outcomes:

1. The formation of the ROH/RBP-TTR complex should be confirmed and purified using gel filtration to separate free TTR and ROH/RBP. Only fractions containing the complex should be used in the experiments. Assuming that the complex is formed with 100% efficiency is overly optimistic.

2. Reloading RBP with isotopically labeled ROH requires an additional purification step. Stripping ROL from the ROH/RBP complex with organic solvent (diethyl ether) is appropriate but relatively harsh, causing partial unfolding of a fraction of RBP. Therefore, assuming that 100% of stripped RBP remains functional and can be reloaded with ROH is inaccurate. Reloading apo-RBP with a stoichiometric amount of ROH without an additional purification step (e.g., ion exchanger) leads to an excess of free ROL and/or its nonspecific association with nonfunctional RBP fractions. Measuring absorbance at 330 nm is not sufficient proof of binding since free ROH also absorbs at the same wavelength.

---

## [Referee Report · Reviewer #3 (Public Review)]

Vitamin A is critical for the development of the brain and for neuronal function and plasticity, however the mechanisms responsible for the uptake of retinol across the blood brain barrier (BBB) are currently not known. The authors investigate vitamin A uptake across the blood brain barrier using an in vitro model based on endothelial cells differentiated from human derived induced pluripotent stem cells. Using recombinant cargo proteins and radioactive tracers the authors then propose a mechanism and a kinetic model for the uptake of retinol across the BBB that requires serum retinol binding protein 4 (RBP4 or RBP) and its receptor stimulated by retinoic acid 6 (STRA6). The results support a concentration dependent mechanism of transport combining a rapid fluid-phase retinol and a slower directed RBP-complexed retinol across the BBB. The data also hint at the potential regulatory roles of TTR on this process independent of its interaction with RBP.

Strengths:

The studies are rigorous and careful and the authors consider free retinol uptake from the fluid-phase in addition to evaluating RBP-TTR and RBP-STRA6 interactions.

The antibody to STRA6 is validated.

The experiments performed are clearly described.

Weaknesses:

The results presented do not offer significant new information regarding the uptake of retinol by tissues beyond what is known and published using genetic, structural and biochemical approaches.

The use of the iPSC-derived BBB model is potentially interesting but this could have been complemented by a thorough genetic dissection of the cellular factors required for the uptake, transcellular transport, and secretion of retinol by the brain endothelial cells.

The conclusions derived are not well supported by the data presented.

It is difficult to infer a mechanism or to derive a meaningful conclusion regarding the in vivo relevance of the results presented.

---

## [Author Response]

We thank the reviewers for their very thorough and detailed comments as well as the overall positive reception of the work. Additionally, the reviewers provided excellent detailed suggestions for future work.

Specific response to Reviewer 1:“Indeed, the major disappointment of this work is the clinical relevance that was highlighted in the Introduction but was never really studied in the end. iPSC from patients could be added to the study.”

We completely agree that it would be very exciting to use patient-derived iPSC in the platform that we describe in this manuscript. We recognize that extensive work to characterize and validate BMECS differentiated from patient-derived iPSCs would be required, including validating BBB-like properties, before retinol transport data could be collected and interpreted. This work is beyond the scope of the current manuscript. We hope that in the future the in vitro model we describe in this manuscript will be used for exactly this type of clinically relevant application.

Specific response to Reviewer 2:“(1) The authors assume that there is a significant fraction of free ROL, 20% for ROH/RBP and 7% for RBP/TTR complexes (summarized in Table 1). This implies that at the physiological concentration of ROH/RBP in the plasma of 2 uM, free ROL represents 0.4 uM. However, the concentration of free ROL is limited by its poor solubility in the aqueous phase, which is around 0.06 uM (Szuts EZ, 1991, Arch Biochem Biophys). Moreover, taking into account the large concentration of other potential nonspecific carriers for lipids, it is safe to assume that there is virtually no free ROH in the plasma. There is also an important physiological reason for the limited amount of free ROL. Its rapid and nonspecific partition into cells (also observed in this study) would work against the highly specific RBP/STRA6-dependent ROH uptake pathway, undermining its physiological function.”

The reviewer raises an important point that we considered carefully during the design of the research. As the reviewer says, Szuts (1991) reported retinol (ROH) solubility of ~0.06 µM (range of 0.03 – 0.11 µM). Szuts defined ROH solubility as ‘the amount of dissolved solute in equilibrium with its solid state…includ[ing] all its dissolved forms (monomers, multimers, and micelles)’. We are using a definition of ‘free’ ROH as ‘ROH not bound to protein’; in our work ‘free’ ROH could include retinol multimers and micelles, which likely do exist under our experimental conditions. (We did not see any evidence of solid ROH.) That said, we calculate that the concentration of free ROH (ROH not bound to protein) is ~0.14 µM when both RBP and TTR are present. In more complex biological mixtures containing other ROH carriers, the concentration of unbound ROH is expected to be lower, in agreement with the reviewer.

One key point is that the free ROH concentration depends on the experimental setup, and must be correctly accounted for. For example, in some of the literature investigating STRA6-mediated uptake and signaling in vitro, purified ROH-RBP is used as the retinol source and samples do not include TTR. In such a case, the unbound ROH concentration in an equilibrated sample is anticipated to be significantly higher than the physiological concentration. Our investigation demonstrates that unbound ROH can accumulate intracellularly; thus, failure to include TTR and/or to account for the action of unbound ROH could lead to errors in mechanistic interpretation of experimental studies on retinol transport into cells or across barriers such as the BBB.

1. “However, a question remains: would the outcome of the experiment be different if the basolateral chamber contained an ROH acceptor (retinol-binding proteins) rather than Hank's balanced salt solution, to which the partition of ROL is limited by its water solubility?”

We agree with the reviewer that it would be very interesting to determine whether retinol permeability changes in the presence of RBP and/or TTR on the basolateral side. This is a logical next step and can readily be performed in the Transwell setup. We chose not to do this for this project because we wanted to compare our setup with other in vitro models (e.g., with porcine BMECs) where no retinol-binding proteins were present basolaterally.

1. “The authors claim that transthyretin (TTR) increases BMECs permeability when compared to ROH/RBP. However, the mechanistic explanation for this phenomenon remains unclear. Do the authors imply the presence of a putative TTR receptor whose signaling could affect the efflux of ROL at the basolateral side of BMECs? TTR is an ubiquitous plasma protein. The concentration of TTR is tightly regulated and maintained between 300 - 330 mg/L. Therefore, it is questionable how TTR can serve as a signaling molecule modulating retinoid homeostasis in the brain.”

We disagree with the reviewer about the TTR concentration. Per Johnson et al (Clin Chem Lab Med 2007, 45:419-426), TTR concentration varies with age, gender, inflammation and nutritional status, with typical concentrations for adults ranging from 150-450 mg/L. We were surprised at our observations that TTR enhanced ROH permeability across BMECs and that LRAT expression increased in the presence of TTR. We do not currently have a mechanistic interpretation and agree with the reviewer that further exploration of these tantalizing observations is warranted.

“Additional technical issues that could affect the experimental outcomes: The formation of the ROH/RBP-TTR complex should be confirmed and purified using gel filtration to separate free TTR and ROH/RBP. Only fractions containing the complex should be used in the experiments. Assuming that the complex is formed with 100% efficiency is overly optimistic.”

We respectfully disagree with the reviewer regarding using gel filtration to isolate TTR/ROH/RBP complexes. Any such isolated complexes will fairly rapidly re-equilibrate so that some protein and some ROH is unbound. It is important to note that we do not assume that the complex is formed with 100% efficiency. In fact, on the contrary, we explicitly take into account the distribution of materials (free TTR, free RBP, free ROH, RBP-ROH, TTR-RBP-ROH) in any sample; values are reported in the manuscript. This issue is also relevant to the first point raised by the reviewer. We routinely validated binding of ROH to RBP by FRET and ROH-RBP to TTR by fluorescence anisotropy.

“Reloading RBP with isotopically labeled ROH requires an additional purification step. Stripping ROL from the ROH/RBP complex with organic solvent (diethyl ether) is appropriate but relatively harsh, causing partial unfolding of a fraction of RBP. Therefore, assuming that 100% of stripped RBP remains functional and can be reloaded with ROH is inaccurate. Reloading apo-RBP with a stoichiometric amount of ROH without an additional purification step (e.g., ion exchanger) leads to an excess of free ROL and/or its nonspecific association with nonfunctional RBP fractions. Measuring absorbance at 330 nm is not sufficient proof of binding since free ROH also absorbs at the same wavelength.”

We produced RBP by refolding of guanidine-denatured RBP in an excess of ROH to ensure near 100% ROH loading. High quality refolded RBP can qualitatively be determined by examination of the A330/280 absorbance ratio, which should be ~1.0. We then extract ROH to completion by diethyl ether to produce pure apo-RBP (ROH-free). We utilized this diethyl-ether stripped apo-RBP stock for all future characterizations, including binding to ROH and TTR. We found our stripped apo-RBP was a suitable replacement for serum sources in every biophysical assay performed. Reloaded ROH-RBP elutes as a single peak on ion exchange chromatography, indicating the vast majority of stripped RBP is available for ROH binding. We provide detailed information about RBP characterization in Est and Murphy, Prot. Exp. Purif. (2020), to which the interested reader is referred.